# Controlling the Inductive Bias of Wide Neural Networks by Modifying the Kernel's Spectrum

**Amnon Geifmann**\*                                                                                     *amnon.geifmann@weizmann.ac.il*
*Weizmann Institute of Science*

**Daniel Barzilai**\*                                                                                     *daniel.barzilai@weizmann.ac.il*
*Weizmann Institute of Science*

**Ronen Basri**                                                                                          *ronen.basri@weizmann.ac.il*
*Weizmann Institute of Science*
*Meta AI*

**Meirav Galun**                                                                                         *meirav.galun@weizmann.ac.il*
*Weizmann Institute of Science*

**Reviewed on OpenReview:** *https: // openreview. net/ forum? id= aD0ExytnEK*

## Abstract

Wide neural networks are *biased* towards learning certain functions, influencing both the rate of convergence of gradient descent (GD) and the functions that are reachable with GD in finite training time. As such, there is a great need for methods that can modify this *bias* according to the task at hand. To that end, we introduce Modified Spectrum Kernels (MSKs), a novel family of constructed kernels that can be used to approximate kernels with desired eigenvalues for which no closed form is known. We leverage the duality between wide neural networks and Neural Tangent Kernels and propose a preconditioned gradient descent method, which alters the trajectory of GD. As a result, this allows for a polynomial and, in some cases, exponential training speedup without changing the final solution. Our method is both computationally efficient and simple to implement.

## 1 Introduction

Recent years have seen remarkable advances in understanding the inductive bias of neural networks. Deep neural networks are biased toward learning certain types of functions. On the one hand, this bias may have a positive effect by inducing an implicit form of regularization. But on the other hand, it also implies that networks may perform poorly when the target function is not well aligned with this bias.

A series of works showed that when the width of a neural network tends to infinity (and with a certain initialization and small learning rate), training a network with GD converges to a kernel regression solution with a kernel called the Neural Tangent Kernel (NTK) (Jacot et al., 2018; Lee et al., 2019; Allen-Zhu et al., 2019; Chizat et al., 2019). Subsequent work showed that the inductive bias of wide neural networks can be characterized by the spectral decomposition of NTK (Arora et al., 2019a; Basri et al., 2019; Yang & Salman, 2019). Following this characterization, we will use the term *spectral bias of neural networks* to refer to the inductive bias induced by their corresponding NTK spectrum. Specifically, it has been observed both theoretically and empirically that for a wide neural network, learning an eigen-direction of the NTK with GD requires a number of iterations that is inversely proportional to the corresponding eigenvalue (Bowman & Montufar, 2022; Fridovich-Keil et al., 2021; Xu et al., 2022). Thus, if this spectral bias can be modified, it could lead to accelerated network training of certain target functions. Typically, the eigenvalue of NTK

---

\*Equal Contribution

decays at least at a polynomial rate, implying that many eigen-directions cannot be learned in polynomial time with gradient descent (Ma & Belkin, 2017). As such, modifying the spectral bias of a neural network is necessary to enable a feasible learning time, allowing learning target functions that are not well aligned with the top eigen-directions of NTK. This prompts the following motivating question:

*Is it possible to manipulate the spectral bias of neural networks?*

The spectrum of a kernel also determines the prediction function when using kernel ridge regression. Therefore, we introduce a family of kernel manipulations that enables generating a wide range of new kernels with nearly arbitrary spectra. These manipulations do not require explicitly working in feature space and are, therefore, computationally efficient. We leverage this technique to design a kernel-based preconditioner that significantly modifies the linear training dynamics of wide neural networks with GD. This preconditioner modifies the spectral bias of the network so that the convergence speed is no longer tied to the NTK but rather to a kernel of our choice. This yields a polynomial or even exponential speedup (depending on the decay rate of the spectrum of the NTK) in the convergence time of GD. We subsequently show that the proposed accelerated convergence does not alter the network's prediction at convergence on test points.

In sum, our main contributions are:

1. We introduce Modified Spectrum Kernels (MSK's), which enable approximating the kernel matrix for many kernels for which a closed form is unknown (Section 3).

2. We introduce preconditioned gradient descent and prove its convergence for wide neural networks (Section 4.1).

3. We prove that our method is consistent, meaning that the preconditioning does not change the final network prediction on test points (Section 4.2).

4. We provide an algorithm that enables a wide range of spectral bias manipulations, yielding a significant convergence speedup for training wide neural networks. We finally demonstrate this acceleration on synthetic data (Section 4.3).

## 2 Preliminaries and Background

We consider positive definite kernels $\boldsymbol{k} : \mathcal{X} \times \mathcal{X} \to \mathbb{R}$ defined over a compact metric space $\mathcal{X}$ endowed with a finite Borel measure $\mu$. Given such $\boldsymbol{k}$, its corresponding (linear) integral operator $L_{\boldsymbol{k}} : L^2_\mu(\mathcal{X}) \to L^2_\mu(\mathcal{X})$ is defined as

$$L_{\boldsymbol{k}}(f)(\mathbf{x}) = \int_{\mathbf{z} \in \mathcal{X}} \boldsymbol{k}(\mathbf{x}, \mathbf{z}) f(\mathbf{z}) d\mu(\mathbf{z}).$$

This is a compact, positive self-adjoint operator that therefore has an eigen-decomposition of the form

$$L_{\boldsymbol{k}}(f) = \sum_{i \geq 0} \lambda_i \langle f, \Phi_i \rangle_\mu \Phi_i,$$

where the inner product $\langle, \rangle_\mu$ is with respect to $L^2_\mu(\mathcal{X})$, and $\lambda_i, \Phi_i$ are the eigenvalues and eigenfunctions of the integral operator satisfying

$$L_{\boldsymbol{k}}(\Phi_i) = \lambda_i \Phi_i.$$

According to Mercer's Theorem, $\boldsymbol{k}$ can be written using the eigen-decomposition of the integral operator as

$$\boldsymbol{k}(\mathbf{x}, \mathbf{z}) = \sum_{i \in I} \lambda_i \Phi_i(\mathbf{x}) \Phi_i(\mathbf{z}), \quad \mathbf{x}, \mathbf{z} \in \mathcal{X}. \tag{1}$$

Furthermore, each such kernel is associated with a unique Reproducing Kernel Hilbert Space (RKHS) $\mathcal{H}_{\boldsymbol{k}} \subseteq L^2_\mu(\mathcal{X})$ (which we denote by $\mathcal{H}$ when the kernel is clear by context), consisting of functions of the

form $f(\mathbf{x}) = \sum_{i \in I} \alpha_i \Phi_i(\mathbf{x})$ whose RKHS norm is finite, i.e., $\|f\|_{\mathcal{H}} := \sum_{i \in I} \frac{\alpha_i^2}{\lambda_i} < \infty$. The latter condition restricts the set of functions in an RKHS, allowing only functions that are sufficiently smooth in accordance to the asymptotic decay of $\lambda_k$. For any positive definite kernel there exists a feature map $\phi : \mathcal{X} \to \mathcal{H}$ s.t $\boldsymbol{k}(\mathbf{x}, \mathbf{z}) = \langle \phi(\mathbf{x}), \phi(\mathbf{z}) \rangle_{\mathcal{H}}$. As such, we may occasionally call the RKHS $\mathcal{H}$ the *feature space* of $\boldsymbol{k}$.

Given training data $X = \{\mathbf{x}_1, ..., \mathbf{x}_n\}$, $\mathbf{x}_i \in \mathcal{X}$, corresponding labels $\{y_i\}_{i=1}^n$, $y_i \in \mathbb{R}$, and a regularization parameter $\gamma > 0$, the problem of *Kernel Ridge Regression (KRR)* is formulated as

$$\min_{f \in \mathcal{H}} \sum_{i=1}^n (f(\mathbf{x}_i) - y_i)^2 + \gamma \|f\|_{\mathcal{H}}. \tag{2}$$

The solution satisfies $f(\mathbf{x}) = \mathbf{k}_{\mathbf{x}}^T (K + \gamma I)^{-1} \mathbf{y}$, where the entries of $\mathbf{k}_{\mathbf{x}} \in \mathbb{R}^n$ are $\frac{1}{n} \boldsymbol{k}(\mathbf{x}, \mathbf{x}_i)$, $K$ is the $n \times n$ kernel matrix with $K_{ij} = \frac{1}{n} \boldsymbol{k}(\mathbf{x}_i, \mathbf{x}_j)$, and $\mathbf{y} = (y_1, ..., y_n)^T$.

## 3 Modified Spectrum Kernels

Our aim in this section is to describe and analyze the construction of novel kernels for which no closed form is known. The novel kernels are constructed by directly manipulating the kernel matrix of existing kernels, which are easy to compute. The theory in this section refers to arbitrary kernels, and the connection to NTK and wide neural networks is deferred to Sec. 4.

**Definition 3.1. Modified Spectrum Kernel (MSK).** Let $\boldsymbol{k}(\mathbf{x}, \mathbf{z}) := \sum_{k=1}^{\infty} \lambda_k \Phi_k(\mathbf{x}) \Phi_k(\mathbf{z})$ be a Mercer kernel with eigenvalues $\lambda_i$ and eigenfunction $\Phi_i(\cdot)$ and $g : \mathbb{R} \to \mathbb{R}$ a function which is non-negative and $L$-Lipschitz. The Modified Spectrum Kernel (w.r.t. $\boldsymbol{k}$) is defined as $\boldsymbol{k}_g(\mathbf{x}, \mathbf{z}) := \sum_{k=1}^{\infty} g(\lambda_k) \Phi_k(\mathbf{x}) \Phi_k(\mathbf{z})$.

The MSK has the same eigenfunctions as the source kernel $\boldsymbol{k}$, while its eigenvalues are modified by $g$. Clearly, constructing a closed-form solution to the modified kernel is rarely possible. However, it is possible to approximate the modified kernel efficiently under certain conditions, given the values of the original kernel $\boldsymbol{k}$, as proved in Theorem 3.2.

**Theorem 3.2.** *Let $g, \boldsymbol{k}, \boldsymbol{k}_g$ be as in Def. (3.1) and assume that $\forall \mathbf{x} \in \mathcal{X}, |\Phi_i(\mathbf{x})| \leq M$. Let $K, K_g$ be the corresponding kernel matrices on i.i.d samples $\mathbf{x}_1, .., \mathbf{x}_n \in \mathcal{X}$. Define the kernel matrix $\tilde{K}_g = V D V^T$ where $V = (\mathbf{v}_1, .., \mathbf{v}_n)$ with $\mathbf{v}_i$ the $i$'th eigenvector of $K$ and $D$ is a diagonal matrix with $D_{ii} = g(\hat{\lambda}_i)$ where $\hat{\lambda}_i$ is the $i$'th eigenvalue of $K$. Then, for $n \to \infty$*

$$\left\| \tilde{K}_g - K_g \right\|_F \overset{a.s.}{\to} 0,$$

*where a.s. stands for almost surely.*

We next provide a proof sketch. The full proof is given in Appendix C. For an eigenfunction $\Phi$ of $\boldsymbol{k}(\mathbf{x}, \mathbf{z})$, we let $\Phi(X) := (\Phi(\mathbf{x}_1), \dots, \Phi(\mathbf{x}_n))^T \in \mathbb{R}^n$ be the evaluation of the eigenfunctions on the data. By the definition of the kernels it can be shown that $K = \sum_{k=1}^{\infty} \lambda_k \Phi_k(X) \Phi_k(X)^T$, and similarly, $K_g = \sum_{k=1}^{\infty} g(\lambda_k) \Phi_k(X) \Phi_k(X)^T$. Since $\tilde{K}_g$ is composed of the eigenvectors of $K$ with the eigenvalues $g(\lambda_k)$, we would like to show that the eigenvalues of $K_g$ are close to $g(\lambda_k)$ and that the eigenvectors of $K$ are close to those of $K_g$. It is already known that the eigenvalues of a kernel matrix converge to those of its integral operator (with the suitable normalization) (Rosasco et al., 2010), and as such, we know that the eigenvalues of $\tilde{K}_g$ should be close to those of $K_g$. The challenge is that the eigenvectors $\mathbf{v}_k$ do not have to be close to $\Phi_k(X)$, the evaluations of the eigenfunctions on the data (for example when there is an eigenvalue of multiplicity greater than 1). This means that the eigenvectors of $\tilde{K}_g$ can be very different from those of $K_g$. We work around this by showing that for large $n$, the eigenvectors of $K$ corresponding to an eigenvalue $\lambda_k$ are related to an eigenfunction $\Phi_i$ as

$$\sum_{\mathbf{v}: K\mathbf{v} = \lambda_k \mathbf{v}} (\mathbf{v}^T \Phi_i(X))^2 \underset{n \to \infty}{\longrightarrow} \begin{cases} 1 & \text{if } \Phi_i \text{ is an eigenfunction of } \lambda_k \\ 0 & \text{else} \end{cases}.$$

This allows us to show that the norm between the eigenspaces of $\tilde{K}_g$ and $K_g$ tends to 0.

**Kernel Construction with MSKs**. Generating new kernels from existing ones is a long-standing problem in kernel theory (Saitoh & Sawano, 2016). The classical kernel theory uses arithmetic operations such as addition and multiplication of kernels to generate new kernels with unique RKHSs. Recent papers provided tools to building data dependent kernels based on training points (Simon, 2022; Sindhwani et al., 2005; Ionescu et al., 2017) . Nevertheless, there are still many kernels for which a closed form is unknown.

MSKs allow extending existing RKHS theory by computing new kernels with predetermined Mercer decomposition even when a closed form is unknown and, specifically, solve KRR with these new kernels. Suppose we would like to solve the problem of KRR as defined in (2). Let $\boldsymbol{k}(\mathbf{x}, \mathbf{z}) = \sum_{k=1}^{\infty} \lambda_k \Phi_k(\mathbf{x})\Phi_k(\mathbf{z})$ and $\boldsymbol{k}_g(\mathbf{x}, \mathbf{z}) = \sum_{k=1}^{\infty} g(\lambda_k)\Phi_k(\mathbf{x})\Phi_k(\mathbf{z})$ be two Mercer kernels where $\boldsymbol{k}(\mathbf{x}, \mathbf{z})$ has a known closed form, whereas $\boldsymbol{k}_g(\mathbf{x}, \mathbf{z})$ does not. Assuming we obtain i.i.d. samples of training points $\mathbf{x}_1, .., \mathbf{x}_n$ and a test point $\mathbf{x}$, we can build $\tilde{K}_g$ (as in Theorem 3.2) using the $n + 1$ points $\mathbf{x}_1, .., \mathbf{x}_n, \mathbf{x}$. Then, by a continuity argument, Theorem 3.2 guarantees that the predictor $f^*(\mathbf{x}) = [\tilde{K}_g]_{n+1,1:n}([\tilde{K}_g]_{1:n,1:n} + \gamma I)^{-1}\mathbf{y}$ converges to the KRR prediction with the kernel $\boldsymbol{k}_g$, where $[\ \cdot\ ]_{:,:}$ corresponds to the sub-matrix induced by the specified indices.

## 4 Provable NTK Based Preconditioning for Neural Networks

In this section, we develop and analyze a preconditioning scheme for accelerating the convergence rate of gradient descent in the training phase of neural networks while minimizing the Mean Squares Error (MSE) loss. The acceleration is achieved by manipulating the spectrum of the NTK, overcoming the spectral bias of neural networks. We begin by explaining how the convergence rate of neural networks is related to the spectrum of the NTK. Then, we introduce a preconditioning scheme for wide neural networks and prove that it attains a global optimum. We further prove that in the infinite width limit and when training the network to completion, preconditioned and standard gradient descent converge to the same global minimizer.

We consider a fully connected neural network parameterized as follows:

$$
\begin{aligned}
\mathbf{g}^{(0)}(\mathbf{x}) &= \mathbf{x} \\
\mathbf{f}^{(l)}(\mathbf{x}) &= W^{(l)}\mathbf{g}^{(l-1)}(\mathbf{x}) + \mathbf{b}^{(l)} \in \mathbb{R}^{d_l}, \qquad l = 1, \dots L \\
\mathbf{g}^{(l)}(\mathbf{x}) &= \rho\left(\mathbf{f}^{(l)}(\mathbf{x})\right) \in \mathbb{R}^{d_l}, \qquad l = 1, \dots L \\
f(\mathbf{x}, \mathbf{w}) &= f^{(L+1)}(\mathbf{x}) = W^{(L+1)} \cdot \mathbf{g}^{(L)}(\mathbf{x}) + b^{(L+1)}.
\end{aligned}
$$

where $\mathbf{w} \in \mathbb{R}^p$ is the set of all the network parameters. We select a simple architecture and note that our results can easily be extended to many other architectures. We denote by $m$ the width of the network and assume that $d_1 = d_2 = .. = d_L = m$. The activation function is denoted by $\rho$ and the following quantities $|\rho(0)|$, $\|\rho'\|_\infty$, and $\sup_{x \neq x'} |\rho'(x) - \rho'(x')|/|x - x'|$ should be finite. The initialization follows the standard NTK parametrization (Lee et al., 2020a) (see more details in Appendix A).

We denote the vector of labels for all the data points $(y_1, \dots, y_n)$ by $\mathbf{y} \in \mathbb{R}^n$ and the vector of network predictions $f(\mathbf{x}_i, \mathbf{w})$ by $f(X, \mathbf{w}) \in \mathbb{R}^n$. The residual at time $t$ is $\mathbf{r}_t := f(X, \mathbf{w}_t) - \mathbf{y}$. Letting $\mathcal{L}$ be the squared error loss, $\mathcal{L}(\mathbf{w}_t) = \frac{1}{2}\|\mathbf{r}_t\|^2$, a gradient descent iteration for optimizing $\mathbf{w} \in \mathbb{R}^p$ is given by

$$
\mathbf{w}_{t+1} - \mathbf{w}_t = -\eta \nabla_{\mathbf{w}}\mathcal{L}(\mathbf{w}_t) = -\eta \nabla_{\mathbf{w}} f(X, \mathbf{w}_t)^T \mathbf{r}_t, \tag{3}
$$

where $\eta$ is the learning rate, and $\nabla_{\mathbf{w}} f(X, \mathbf{w}_t) \in \mathbb{R}^{n \times p}$ is the Jacobian of the network.

The empirical NTK matrix at time $t$, $K_t \in \mathbb{R}^{n \times n}$, and the NTK matrix, $K \in \mathbb{R}^{n \times n}$, are defined as

$$
\begin{aligned}
K_t &= \frac{1}{m} \nabla_{\mathbf{w}} f(X, \mathbf{w}_t) \nabla_{\mathbf{w}} f(X, \mathbf{w}_t)^T \tag{4} \\
K &= \lim_{m \to \infty} K_0. \tag{5}
\end{aligned}
$$

We assume that $\lambda_{\min}(K) > 0$. A simple case where this condition is satisfied is whenever $\mathcal{X} = \mathbb{S}^{d-1}$ and $\rho$ grows non-polynomially (e.g ReLU) Jacot et al. (2018), and also various other settings as given by Oymak &

Soltanolkotabi (2020); Wang & Zhu (2021); Nguyen et al. (2021); Montanari & Zhong (2022); Barzilai & Shamir (2023).

Arora et al. (2019a) showed that for a wide neural network with ReLU activation, small initialization, and a small learning rate, it holds that for every $\epsilon > 0$, the residual at time $t$ evolves with the following linear dynamics

$$\|\mathbf{r}_t\| = \|(I - \eta K)^t \mathbf{y}\| \pm \epsilon = \sqrt{\sum_{i=1}^{n}(1 - \eta\lambda_i)^{2t}(\mathbf{v}_i^T \mathbf{y})^2} \pm \epsilon, \tag{6}$$

where $\{\lambda_i\}_{i=1}^{n}$ and $\{\mathbf{v}_i\}_{i=1}^{n}$ respectively are the eigenvalues and eigenvectors of the NTK matrix $K$.

Eq. (6) reveals the relation between the inductive bias of neural networks and the spectrum of the NTK matrix (Basri et al., 2020b). Specifically, to learn an eigen-direction $\mathbf{v}_i$ of a target function within accuracy $\delta > 0$, it is required that $(1 - \eta\lambda_i)^t < \delta + \epsilon$. When the learning rate is sufficiently small to imply convergence, $0 < \eta < \frac{2}{\lambda_1}$, the number of iterations needed is

$$t > -\log(\delta + \epsilon)/\eta\lambda_i = O\left(\frac{\lambda_1}{\lambda_i}\right).$$

The eigenvalues and eigenvectors of $K$ depend on the data distribution and are not yet fully characterized for arbitrary distributions (e.g., Basri et al. (2020a)). In the case of a fully connected network with ReLU activation and with data points distributed uniformly on the sphere $\mathbb{S}^{d-1}$, the eigenvectors are discretizations of the spherical harmonics and the eigenvalues asymptotically decay as $\lambda_k \approx k^{-d}$, where $k$ is the frequency (Basri et al., 2020b; Bietti & Bach, 2020). In this scenario, learning a high-frequency eigenvector with gradient descent is computationally prohibitive, even for a low-dimension sphere. With other activation functions, the asymptotic decay of the eigenvalues might be even exponential (Murray et al., 2022), yielding an infeasible computational learning cost for learning high frequency target functions.

### 4.1 Preconditioned Gradient Descent

To accelerate the convergence of standard gradient descent (3), we propose a *preconditioned gradient descent* (PGD). The update rule of our PGD is given by

$$\mathbf{w}_{t+1} = \mathbf{w}_t - \eta\nabla_\mathbf{w} f(X, \mathbf{w}_t)^T S\mathbf{r}_t, \tag{7}$$

where $S \in \mathbb{R}^{n \times n}$ is a preconditioning matrix that satisfies $S \succ 0$.

Standard preconditioning techniques multiply the network's gradient from the left by a $p \times p$ matrix (where $p$ is the number of parameters in the network, usually huge). In contrast, our preconditioner multiplies the network's gradient from the right by an $n \times n$ matrix, reducing the cost per iteration from $p^2$ to $n^2$. This is significant since in the over-parameterized case $n \ll p$.

We next derive the linear dynamics of the form of (6) for PGD. One of the key properties of PGD, is that carefully choosing $S$ allows modifying the dynamics of gradient descent almost arbitrarily.

**Theorem 4.1.** *Suppose assumptions 1-4 are satisfied. Let $\eta_0, \delta_0, \epsilon > 0$, $S \in \mathbb{R}^{n \times n}$ such that $S \succ 0$ and $\eta_0 < \frac{2}{\lambda_{min}(KS)+\lambda_{max}(KS)}$. Then, there exists $N \in \mathbb{N}$ such that for every $m \geq N$, the following holds with probability at least $1 - \delta_0$ over random initialization when applying preconditioned GD with learning rate $\eta = \eta_0/m$*

$$\mathbf{r}_t = (I - \eta_0 KS)^t\mathbf{y} \pm \xi(t),$$

*where $\|\xi\|_2 \leq \epsilon$.*

Recall that we assume $K \succ 0$ and $S \succ 0$, so while $KS$ is not necessarily positive definite or symmetric, it has positive real eigenvalues. As such, the term $\frac{2}{\lambda_{min}(KS)+\lambda_{max}(KS)}$ is positive and defines the maximal feasible learning rate. The formal proof of Theorem 4.1 is given in Appendix A, and here we give some key points

of the proof. The proof relies on Lemma 4.2, which shows that PGD finds a global minimum in which the weights are close to their initial values. In particular, for any iteration $t$, $K_t S \approx K_0 S \approx KS$. Based on the results of Lemma 4.2, Theorem 4.1 carefully bounds the distance $\xi(t)$ between $\mathbf{r}_t$ and the linear dynamics $(I - \eta_0 KS)^t \mathbf{y}$.

**Lemma 4.2.** *Suppose assumptions 1-4 are satisfied. For $\delta_0 > 0$, $\frac{2}{\lambda_{min}(KS) + \lambda_{max}(KS)} > \eta_0$ and $S$ such that $S \succ 0$, there exist $C > 0$, $N \in \mathbb{N}$ and $\kappa > 1$, such that for every $m \geq N$, the following holds with probability at least $1 - \delta_0$ over random initialization. When applying preconditioned gradient descent as in (7) with learning rate $\eta = \eta_0/m$*

1. $\|\mathbf{r}_t\|_2 \leq \left(1 - \frac{\eta \lambda_{min}}{3}\right)^t C$

2. $\sum_{j=1}^{t} \|\mathbf{w}_j - \mathbf{w}_{j-1}\|_2 \leq \frac{3\kappa C}{\lambda_{min}} m^{-1/2}$

3. $\sup_t \|(K_0 - K_t)S\|_F \leq \frac{6\kappa^3 C}{\lambda_{min}} m^{-1/2}$,

*where $\lambda_{min}$ is the minimal eigenvalue of $KS$.*

The full proof of Lemma 4.2 is given in Appendix A.

Theorem 4.1 implies that the dynamics of preconditioned gradient descent are characterized by $KS$ instead of $K$. In particular, if $KS$ is symmetric, PGD follows the dynamics

$$\|\mathbf{r}_t\| = \left\|(I - \eta KS)^t \mathbf{y}\right\| \pm \epsilon = \sqrt{\sum_{i=1}^{n} (1 - \eta \hat{\lambda}_i)^{2t} (\hat{\mathbf{v}}_i^T \mathbf{y})^2} \pm \epsilon, \tag{8}$$

where now, in contrast to (6), $\hat{\lambda}_i, \hat{\mathbf{v}}_i$ are the eigenvalues and eigenvectors of $KS$. The role of the preconditioner $S$ is to reduce the condition number of $K$, i.e., improving the worst-case convergence rate. Moreover, it can be chosen to accelerate the convergence rate in directions corresponding to any eigenvector of the NTK matrix.

The results of Zhang et al. (2019) and Cai et al. (2019) can be viewed as a special case of (8). Specifically, these works characterize the convergence of an approximate second-order method (Gauss-Newton) when applied to a wide, two-layer network. In this case, the update rule is of the form

$$\mathbf{w}_{t+1} - \mathbf{w}_t = -\eta F(\mathbf{w}_t)^{-1} \nabla_{\mathbf{w}} \mathcal{L}(\mathbf{w}_t),$$

with $F$ being the Fisher information matrix associated with the network's predictive distribution. Their derivation leads to similar convergence results as shown in (8) but is limited to the case of $S = K_t^{-1}$. For sufficiently wide networks, $K_t$ is very close to $K$ (in spectral norm). Thus, taking $S = K^{-1}$ suffices for achieving the same convergence rates as if $S = K_t^{-1}$. Furthermore, our method requires computing the preconditioning matrix only once and is therefore more efficient. In fact, Figure 1 demonstrates that taking $S = K^{-1}$ leads to an even faster convergence.

We show in Sec. 4.3 how $S$ can be chosen to reduce by a polynomial (or even exponential) factor the number of iterations required to converge in the directions corresponding to small eigenvalues of $K$. This is highly beneficial when the projections of the target function on the eigenvectors of $K$, $(\hat{\mathbf{v}}_i^T \mathbf{y})$ are relatively large in directions corresponding to small eigenvalues. For example, for image reconstruction or rendering, this equates to learning more efficiently the fine details, which correspond to high frequencies in the image (Tancik et al., 2020). Another example arises in Physics Informed Neural Networks (PINN), which were shown to suffer significantly from the slow convergence rate in the directions corresponding to small eigenvalues (Wang et al., 2022).

## 4.2 Convergence Analysis

In this section, we characterize the resulting prediction function of the network on unseen test points and show that PGD generates a consistent prediction function in the following sense. At convergence, the prediction

function of a network trained with PGD is close to the prediction function of a network trained with standard GD.

We start by defining a preconditioned loss, $\mathcal{L}_S(f, \mathbf{w})$, which modifies the standard MSE loss

$$\mathcal{L}_S(f, \mathbf{w}) = \frac{1}{2} \left\| S^{1/2}(f(X, \mathbf{w}) - \mathbf{y}) \right\|_2^2, \tag{9}$$

where $S \succ 0$ is the preconditioning matrix. An iteration of standard GD w.r.t $\mathcal{L}_S(f, \mathbf{w})$ yields

$$\mathbf{w}_{t+1} - \mathbf{w}_t = -\eta \nabla_{\mathbf{w}} \mathcal{L}_S(f, \mathbf{w}_t) = -\eta \nabla_{\mathbf{w}} f(X, \mathbf{w}_t)^T S \mathbf{r}_t, \tag{10}$$

which is equivalent to a PGD iteration (7) with the standard MSE loss. This has two implications. First, it implies that PGD can easily be implemented by simply modifying the loss function. Second, it enables us to analyze the prediction function generated by PGD.

Specifically, given an initialization $\mathbf{w}_0 \in \mathbb{R}^p$, let $\phi(\mathbf{x}) := \nabla f(\mathbf{x}, \mathbf{w}_0)$ and $h(\mathbf{x}, \mathbf{w}') := \langle \mathbf{w}', \phi(\mathbf{x}) \rangle$. For simplicity, we denote $h(X, \mathbf{w}') := (h(\mathbf{x}_1, \mathbf{w}'), \dots, h(\mathbf{x}_n, \mathbf{w}'))^T \in \mathbb{R}^n$.

The minimizer of the kernel ridge regression objective with respect to the preconditioned loss is defined as

$$\mathbf{w}_\gamma^* := \arg\min_{\mathbf{w}' \in \mathbb{R}^p} \frac{1}{2} \left\| S^{1/2}(h(X, \mathbf{w}') - \mathbf{y}) \right\|_2^2 + \frac{1}{2}\gamma \left\| \mathbf{w}' \right\|_2^2, \tag{11}$$

and the minimizer of a kernel ridge regression with respect to the standard MSE loss is defined as

$$\mathbf{w}_\gamma^{**} := \arg\min_{\mathbf{w}' \in \mathbb{R}^p} \frac{1}{2} \left\| (h(X, \mathbf{w}') - \mathbf{y}) \right\|_2^2 + \frac{1}{2}\gamma \left\| \mathbf{w}' \right\|_2^2. \tag{12}$$

We denote by $\mathbf{w}^*$ and $\mathbf{w}^{**}$ the limits of $\mathbf{w}_\gamma^*$ and $\mathbf{w}_\gamma^{**}$, respectively, when $\gamma \to 0$. In Lemma 4.3 we show that the two minimizers are equal, which means that the preconditioned loss does not change the prediction function in kernel regression.

For a given learning rate $\eta$ and loss $\mathcal{L}_S$, $h$ can be optimized through gradient descent with respect to the loss $\mathcal{L}_S$ via the iterations

$$\mathbf{w}_0' = 0 \quad ; \quad \mathbf{w}_{t+1}' = \mathbf{w}_t' - \eta \nabla \mathcal{L}_S(h, \mathbf{w}_t'). \tag{13}$$

We show in Lemma 4.3 that this iterative procedure yields a prediction function that remains close to the neural network prediction throughout PGD.

**Lemma 4.3.**      *1. Let $\mathbf{w}_0 \in \mathbb{R}^p$ and $\phi(\mathbf{x}) := \nabla f(\mathbf{x}, \mathbf{w}_0)$, it holds that $\mathbf{w}^* = \mathbf{w}^{**}$.*

     *2. Let $\delta_0 > 0$, $\epsilon > 0$, a test point $\mathbf{x} \in \mathbb{R}^d$ and $T > 0$ number of iteration. Under the conditions of Theorem 4.1, $\exists N \in \mathbb{N}$ s.t $\forall m > N$, it holds with probability at least $1 - \delta_0$ that*

$$|h(\mathbf{x}, \mathbf{w}_T') - f(\mathbf{x}, \mathbf{w}_T)| \leq \epsilon,$$

     *where $\mathbf{w}'$ is as in (13) and $f$ is optimized with PGD.*

The full proof of Lemma 4.3 can be found in Appendix B. Point (1) is proved by deriving $\mathbf{w}_\gamma^*$ in closed form, i.e., $\mathbf{w}_\gamma^* = \sum_{i=1}^n \alpha_i^* \phi(\mathbf{x}_i)$, with

$$\alpha^* = (K_0 + \gamma S^{-1})^{-1} \mathbf{y}.$$

This serves as a generalization of the standard kernel ridge regression solution, which is obtained by substituting $S = I$. Therefore, using the preconditioned loss yields a corresponding change of the regularization. When taking $\gamma \to 0$, the solution becomes equivalent to that of the standard kernel ridge regression problem.

Point (2) states that in the limit of infinite width, the value of $f(\cdot, \mathbf{w}_T)$ at a test point is very close to $h(\cdot, \mathbf{w}_T')$. We note that as $\mathcal{L}_S$ is convex, $h(\mathbf{x}, \mathbf{w}_T')$ approaches $h(\mathbf{x}, \mathbf{w}^*)$ for sufficiently large $T$.

By combining these two points we obtain that, with sufficiently many iterations and large width, $f$ well approximates the solution of the standard kernel ridge regression problem, and thus of the standard non-preconditioned gradient descent on the network itself (Lee et al., 2019).

---

**Algorithm 1** Preconditioned Gradient descent

---

**Require:** $X, \mathbf{y}, f(\mathbf{x}, \mathbf{w}), K, \epsilon, g(\cdot), w_0$
    Decompose $K \leftarrow V^T D V$
    Define $S \leftarrow I - \sum_{i=1}^{k} \left(1 - \frac{g(\lambda_i)}{\lambda_i}\right) \mathbf{v}_i \mathbf{v}_i^T$
    $t \leftarrow 0$
    **while** $\|\mathbf{r}_t\| > \epsilon$ **do**
        $\mathbf{w}_{t+1} \leftarrow \mathbf{w}_t - \eta \nabla_{\mathbf{w}} f(X, \mathbf{w}_t)^T S \mathbf{r}_t$
        $t \leftarrow t + 1$
    **end while**

---

### 4.3 An Algorithm for Modifying the Spectral Bias

In light of Theorem 4.1, $S$ can be chosen to mitigate any negative effect arising from NTK. Specifically, we can modify the spectral bias of neural networks in a way that enables them to efficiently learn the eigen-directions of the NTK that correspond to small eigenvalues. To this end, we use a MSK based precondition, as outlined in the PGD algorithm 1.

First, given the NTK matrix $K$ of size $n \times n$, we construct a pre-conditioner from $K$ by applying a spectral decomposition, obtaining the top $k + 1$ eigenvalues and eigenvectors. We denote the top $k + 1$ eigenvalues by $\lambda_1 \geq \ldots \geq \lambda_k \geq \lambda_{k+1} > 0$, and their corresponding eigenvectors by $\mathbf{v}_1, \ldots, \mathbf{v}_k, \mathbf{v}_{k+1}$. The proposed pre-conditioner is of the form

$$S = I - \sum_{i=1}^{k} \left(1 - \frac{g(\lambda_i)}{\lambda_i}\right) \mathbf{v}_i \mathbf{v}_i^T. \tag{14}$$

It can be readily observed that $S$ and $K$ share the same eigenvectors, and as long as $g(\lambda_i) > 0$ then $S \succ 0$ and its eigenvalues given in descending order are:

$$\left[1, \ldots, 1, \frac{g(\lambda_k)}{\lambda_k}, \ldots, \frac{g(\lambda_1)}{\lambda_1}\right].$$

Since $S$ and $K$ share the same eigenvectors, it holds that the eigenvalues of $KS$ are

$$[g(\lambda_1), \ldots, g(\lambda_k), \lambda_{k+1}, \lambda_{k+2}, .., \lambda_n].$$

The connection between the proposed preconditioner and Theorem 3.2 can now be clearly seen, as the product $KS$ yields a MSK matrix, and therefore approximates a kernel that shares the same eigenfunctions of $\mathbf{k}$ and its eigenvalues are modified by $g$ (where for $\lambda \geq \lambda_{k+1}$ we let $g(\lambda) = \lambda$). Together with Theorem 4.1, the dynamics of the neural network are controlled by this modified kernel, and as such we obtain a way to alter the spectral bias of neural networks. From Eq. (8), we obtain the following corollary, stating that the top $k + 1$ eigenvectors $\mathbf{v}_i$ of $K$ can be learnt in $O(1)$ time.

**Corollary 4.4.** *Let $g(\lambda_i) = \lambda_{k+1}$, $1 \leq i \leq k$, and $S$ be given by (14) (making the top $k + 1$ eigenvalues of $KS$ equal to each other). Under the conditions of Theorem 4.1, by picking the learning rate $\eta = \frac{2}{\lambda_{k+1}+\lambda_n}$, $\exists N \in \mathbb{N}$ s.t $\forall m > N$, it holds with probability at least $1 - \delta_0$ that for every $i \leq k + 1$, $\mathbf{v}_i$ can be learnt (up to $\epsilon$ error) in $O(1)$ time (number of iterations).*

This starkly contrasts the typical spectral bias of neural networks, under which for large $i$, learning $\mathbf{v}_i$ may be infeasible in polynomial time. For example, when the activation function $\rho$ is tanh, Murray et al. (2022) showed that $\lambda_k = O\left(k^{-d+1/2}e^{-\sqrt{k}}\right)$. Applying (6), vanilla GD requires $O\left(k^{d+1/2}e^{\sqrt{k}}\right)$ iterations to converge, exponentially slower in $k$ compared to PGD.

Another natural choice for $g$, is given some $\tilde{\gamma} > 0$, to let $g(\lambda) := \lambda + \tilde{\gamma}$ and $k = n$, implying that $KS = K + \tilde{\gamma}I$. Thus, we obtain a direct correspondence between PGD with such a choice of $g$, and *regularized* kernel regression. Such a choice of $g$ may help prevent overfitting, even without early stopping. This is demonstrated in Figure 2.

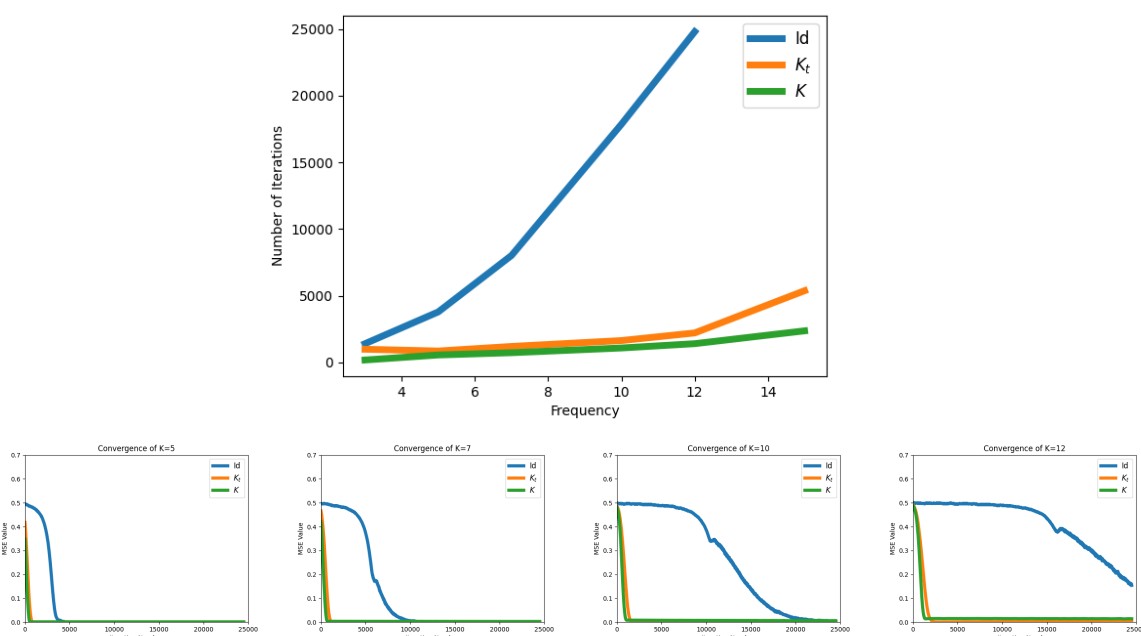

Figure 1: Numerical validation. Top: The number of iterations required to learn different Fourier components as a function of frequency. Standard SGD is shown in blue, and (stochastic) PGD with a preconditioner derived from the NTK matrix $K$ and the empirical NTK $K_t$ respectively are shown in green and orange. Bottom: Training curves with four different frequencies ($k = 5, 7, 10, 12$). The graphs show the MSE loss as a function of iteration number with stochastic GD and PGD.

The NTK matrix may be ill-conditioned with the smallest eigenvalues very close to 0. This issue is magnified by the fact that some approximation may be needed when computing the NTK for arbitrary architectures Novak et al. (2022); Mohamadi et al. (2023). However, a small $k$ can help avoid numerical instabilities in the preconditioner by only modifying sufficiently large eigenvalues. Furthermore, since only the top $k$ eigenvalues and eigenvectors need to be calculated, choosing a small $k$ allows for a more efficient calculation of $S$. By contrast, choosing $k = n$ implies that $S = K^{-1}$. We leave the choice of $k$, the number of modified eigenvalues, to the practitioners, but in general, one should think of $k$ as providing a trade-off between the worst-case rate of convergence, and computational stability and efficiency.

Although our method involves computing the preconditioner, it still results in more efficient optimization in the over-parameterized regime. To see this, we note that for $L > 2$, if the width $m$ of every layer is at least $\sim n^q$ for some $q > 0$, then the number of parameters of a fully-connected network is $\Omega(n^{2q})$. This implies that the worst-case number of operations for GD to converge is $\omega\left(\frac{n^{2q}}{\lambda_{min}(K)}\right)$ where $\lambda_{min}(K)$ is the minimal eigenvalue of $K$. For a fully connected ReLU network, upper bounds on $\lambda_{\min}(K)$ decay polynomially in $n$ (Barzilai & Shamir, 2023), implying that the complexity is $\omega(n^{2q+1})$. Under the most general assumptions, Song & Yang (2019) achieved $q = 4$. In this case, we have a clear computational advantage since inverting $K$ or calculating its eigenvalues costs $O(n^3)$. Nevertheless, under stricter assumptions, smaller widths than $q = 4$ may suffice, but our method is still more efficient even for linear width ($q = 1$). Our method is also significantly more efficient than a standard preconditioner matrix whose size is quadratic in the number of parameters. Furthermore, our preconditioner needs to be computed only once, so its computational complexity is unrelated to the number of iterations needed to converge, which as already discussed, can be exponential. Lastly, in practice, one may choose $k$, the number of eigenvalues to modify, to be small, leading to further speedups.

**Experiments**. We validated our results numerically by tracking the speed of convergence for a neural network both with and without a preconditioner. We generated inputs from a uniform distribution on the circle,

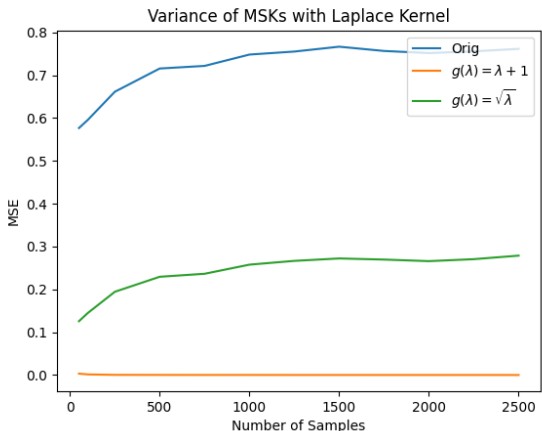

Figure 2: The effect of various MSKs on overfitting to noise. Choosing an MSK with a slower eigenvalue decay helps prevent overfitting. Starting from a Laplace kernel, for each choice of $g$, we perform unregularized kernel regression (with $\gamma = 0$) using an MSK as defined in 3.1. The target function is identically 0, with i.i.d Gaussian noise added to the training set ($y_i \sim \mathcal{N}(0, 1)$). The $x$ axis denotes the number of samples, and $y$ axis the MSE on the test set (for which the target is 0).

$\mathbf{x}_i \sim U(\mathbb{S}^1)$. With each input we associated a target value $y_i$ by taking a Fourier component of frequency $k$ (we repeated this experiment with different values of $k$), i.e., if $\mathbf{x}_i = (\cos(\theta), \sin(\theta))^T$ then $y_i = \sin(k\theta)$. Note that under the uniform distribution on the circle, the Fourier components are the eigenfunctions of the integral operator of NTK for fully connected networks (Basri et al., 2019). We then trained a fully connected network to regress the target function. For efficiency, we trained the network with stochastic gradient descent (SGD). The preconditioner $S$ is chosen to be either $I_n$ (no preconditioning), $K^{-1}$ (inverse NTK matrix), or $K_t^{-1}$ (inverse empirical NTK matrix). Figure 1 shows that without preconditioning, the number of standard GD iterations is roughly quadratic in the frequency of the target function and quickly reaches the iteration limit. In contrast, using preconditioners based on either $K$ or $K_t$ boosts the convergence, yielding a near-constant number of iterations, irrespective of the frequency of the target function. It can be seen that without preconditioning, the number of standard GD iterations is roughly quadratic in the frequency of the target function and quickly reaches the iteration limit. In contrast, using preconditioners based on either $K$ or $K_t$ boosts the convergence, yielding a near constant number of iterations, irrespective of the frequency of the target function. The lower panel in Figure 1 also shows a convergence plot for a few different functions.

Furthermore, we analyze the *variance* of kernel regression under different choices of MSKs in Figure 2. The error of kernel regression is often decomposed into a bias and variance term, where the variance is the error when training on labels given by a constant 0 target function with noise added to the training set (Tsigler & Bartlett, 2023). In our experiment, starting from a Laplace kernel, for each choice of $g$, we perform unregularized kernel regression (with $\gamma = 0$) using an MSK as defined in 3.1. We take inputs drawn uniformly from $\mathbb{S}^2$ and $y_i \sim \mathcal{N}(0, 1)$. The $y$ axis depicts the MSE over a test set of 1000 samples (where the test set target labels are 0), averaged over 25 trials. Various works suggest that a slower eigenvalue decay helps induce an implicit regularization in the kernel, helping it to avoid overfitting (Mallinar et al., 2022; Tsigler & Bartlett, 2023). We can observe this phenomenon in Figure 2, where the MSKs with a slower eigenvalue decay achieve a smaller variance.

## 5   Related Works

A long string of works established the connection between kernel regression and neural networks. Early works considered a Bayesian inference setting for neural networks, showing that randomly initialized networks are equivalent to Gaussian processes (Williams, 1997; Neal, 2012). In this direction, Cho & Saul (2009)

introduced the Arc-cosine kernel, while Daniely et al. (2016) studied general compositional kernels and its relation to neural networks. Recently, a family of kernels called the Neural Tangent Kernels (NTK) was introduced (Jacot et al., 2018; Allen-Zhu et al., 2019; Arora et al., 2019b). These papers proved that training an over-paramterized neural network is equivalent to using gradient decent to solve kernel regression with NTK. Follow-up works defined analogous kernels for residual networks (Huang et al., 2020), convolutional networks (Arora et al., 2019b; Li et al., 2019), and other standard architectures (Yang, 2020).

Subsequent work used the NTK theory to characterize the spectral bias of neural networks. Bach (2017); Basri et al. (2019); Cao et al. (2019); Bietti & Bach (2020); Xu et al. (2019) have studied the eigenfunctions and eigenvalues of NTK under the uniform distribution and further showed that fully connected neural networks learn low-frequency functions faster than higher-frequency ones. Basri et al. (2020a) further derived the eigenfunctions of NTK with non-uniform data distribution. Yang & Salman (2019); Misiakiewicz & Mei (2021) analyzed the eigenvalues of NTK over the Boolean cube. More recent studies investigated this bias with other network architectures, including fully-connected ResNets (Belfer et al., 2021; Tirer et al., 2022), convolutional networks (Geifman et al., 2022; Cagnetta et al., 2022; Xiao, 2022), and convolutional ResNets (Barzilai et al., 2022). (Murray et al., 2022) characterized the spectrum of NTK using its power series expansion.

Preconditioning is a widely used approach to accelerating convex optimization problems. A preconditioner is typically constructed using the inverse of the Hessian or its approximation (Nocedal & Wright, 1999) to improve the condition number of the problem. Recent works tried to improve the condition number of kernel ridge regression. Ma & Belkin (2017); Ma et al. (2018) suggested using a left preconditioner in conjunction with Richardson iterations. Another line of work analyzed the speed of convergence of an approximate second order method for two-layer neural networks (Zhang et al., 2019; Cai et al., 2019). They showed that natural gradient descent improves convergence in a factor proportional to the condition number of the NTK Gram matrix.

Several studies aim to construct preconditioners for neural networks. Carlson et al. (2015) built a diagonal preconditioner based on Adagrad and RMSProp updates. Other heuristic precondtioiners use layer-wise adaptive learning rates (You et al., 2017; 2019). In this line of work, it is worth mentioning (Lee et al., 2020b), who used the convergence results of Arora et al. (2019b) to incorporate leverage score sampling into training neural networks. Amari et al. (2020) studied the generalization of overparameterized ridgeless regression under a general class of preconditioners via the bias-variance trade-off.

Past studies explored the concept of data-dependent kernels. Simon (2022) suggested using the posterior of the target function to produce a new type of data-dependent kernel. Sindhwani et al. (2005) studied a wide class of kernel modifications conditioned on the training data and explored their RKHS. Ionescu et al. (2017) extended this method to build a new class of kernels defined by multiplying the random feature approximation with a weighted covariance matrix. Kennedy et al. (2013) constructed a new family of kernels for a variety of data distributions using the Spherical Harmonic functions on the 2-Sphere.

## 6  Conclusion

In this work we addressed the problem of manipulating the spectral bias of neural networks. We formulated a unique preconditioning scheme which enables manipulating the speed of convergence of gradient descent in each eigen-direction of the NTK. This preconditioning scheme can be efficiently implemented by a slight modification to the loss function. Furthermore, for sufficient training time and width, we showed that the predictor obtained by standard GD is approximately the same as that of PGD. We also showed how to construct novel kernels with nearly arbitrary eigenvalues through the use of kernel spectrum manipulations. Our theory is supported by experiments on synthetic data. In future work we plan to explore other forms of spectral manipulations and apply our method in real-world scenarios.

**Acknowledgments**

Research at the Weizmann Institute was partially supported by the Israel Science Foundation, grant No. 1639/19, by the Israeli Council for Higher Education (CHE) via the Weizmann Data Science Research Center,

by the MBZUAI-WIS Joint Program for Artificial Intelligence Research and by research grants from the Estates of Tully and Michele Plesser and the Anita James Rosen and Harry Schutzman Foundations.

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

# A  Convergence of PGD

In this section we prove Theorem 4.1. We begin with assumptions and notations for the proof. The assumptions follows those of Lee et al. (2019):

1. The width of the hidden layers are identical and equal to $m$

2. The analytic NTK $K$ is full rank with $\lambda_{min}(K) > 0$.

3. The training set $\{\mathbf{x}_i, y_i\}_{i=1}^n$ is contained in some compact set.

4. The activation function $\rho$ satisfies

$$|\rho(0)|, \ \|\rho'\|_\infty, \ sup_{x \neq x'}|\rho'(x) - \rho'(x')|/|x - x'| < \infty$$

The network parametrization is as follows

$$
\begin{aligned}
\mathbf{g}^{(0)}(\mathbf{x}) &= \mathbf{x} \\
\mathbf{f}^{(l)}(\mathbf{x}) &= W^{(l)}\mathbf{g}^{(l-1)}(\mathbf{x}) + \beta\mathbf{b}^{(l)} \in \mathbb{R}^{d_l}, \qquad l = 1, \dots L \\
\mathbf{g}^{(l)}(\mathbf{x}) &= \rho\left(\mathbf{f}^{(l)}(\mathbf{x})\right) \in \mathbb{R}^{d_l}, \qquad l = 1, \dots L \\
f(\mathbf{x}, \mathbf{w}) &= f^{(L+1)}(\mathbf{x}) = W^{(L+1)} \cdot \mathbf{g}^{(L)}(\mathbf{x}) + b^{(L+1)}
\end{aligned}
$$

The network parameters $\mathbf{w}$ include $W^{(L+1)}, W^{(L)}, ..., W^{(1)}$, where $W^{(l)} \in \mathbb{R}^{d_l \times d_{l-1}}$, $\mathbf{b}^{(l)} \in \mathbb{R}^{d_l \times 1}$, $W^{(L+1)} \in \mathbb{R}^{1 \times d_L}$, $b^{(L+1)} \in \mathbb{R}$, $\rho$ is the activation function. The network parameters are initialized with $\mathcal{N}(0, \frac{c_\rho}{d_l})$ for $c_\rho = 1/\left(\mathbb{E}_{z \sim \mathcal{N}(0,1)}[\rho(z)^2]\right)$, except for the biases $\{\mathbf{b}^{(1)}, \dots, \mathbf{b}^{(L)}, b^{(L+1)}\}$, which are initialized with $\mathcal{N}(0, c_\rho)$. We further set at the last layer $c_\rho = \nu$ for a small constant $\nu$.

For the ease of notation we use the following short-hands:

$$f(\mathbf{w}_t) = f(X, \mathbf{w}_t) \in \mathbb{R}^n \tag{15}$$

$$\mathbf{r}_t = f(X, \mathbf{w}_t) - Y \in \mathbb{R}^n \tag{16}$$

$$J(\mathbf{w}_t) = \nabla f(X, \mathbf{w}_t) \in \mathbb{R}^{n \times p} \tag{17}$$

The empirical and analytical NTK is defined as

$$
\begin{aligned}
K_t &:= \frac{1}{m}J(\mathbf{w}_t)J(\mathbf{w}_t)^T \\
K &:= lim_{m \to \infty}K_0
\end{aligned}
$$

Since $f(X, \mathbf{w}_0)$ converge in distribution to gaussian process with zero mean, we denote by $R_0$ the constant such that with probability at least $(1 - \delta_0)$ over the random initialization

$$\|\mathbf{r}_t\| \leq R_0 \tag{18}$$

Likewise we have that with the same probability $\|f(X, \mathbf{w}_0)\| \leq \nu C_0$ for a constant $C_0$.

Finally, our PGD updates are

$$\mathbf{w}_{t+1} = \mathbf{w}_t - \eta J(\mathbf{w}_t)^T S\mathbf{r}_t \tag{19}$$

We let $\lambda_{\min} := \lambda_{\min}(KS)$. Our proofs are carried out for a fixed preconditioner $S$ which does not depend on $t$. However, the same proofs would hold for $S_t$ which changes with $t$, if one makes the additional assumption that $\|S_t\|_F$ is uniformly bounded for all $t$. Of course, if $S$ is fixed, then $\|S\|_F$ is finite and thus trivially bounded. This is used in Lemma $A.1$ and we will make its use clear.

The bounded dataset of Assumption 3 is used for simplicity in two places and can in both cases, be weakened. The assumption is first used to ensure that the kernel has a Mercer decomposition, as in Eq. (1). Weaker

assumptions (which are slightly more complicated) can be used in this case (Steinwart & Scovel, 2012). For example, one has a Mercer decomposition if $X$ is a Hausdorff space (any metric space suffices), the kernel function is continuous, $\mu$ is a probability measure, and the function $h(x) := k(x, x)$ has finite L2 norm. The boundedness assumption is also used to bound $f$ at initialization. However, even for many unbounded distributions, $f$ can be bounded with high probability. As such, both requirements are satisfied by many unbounded distributions, such as Gaussians, and our statements can thus be naturally extended to many such unbounded distributions.

**Lemma A.1.** *There is a $\kappa > 0$ such that for every $C > 0$, with high probability over random initialization the following holds $\forall \mathbf{w}, \tilde{\mathbf{w}} \in B(\mathbf{w}_0, Cm^{-1/2})$:*

*1.* $\frac{1}{\sqrt{m}} \left\| (J(\mathbf{w}) - J(\tilde{\mathbf{w}}))S \right\|_F \leq \kappa \left\| \mathbf{w} - \tilde{\mathbf{w}} \right\|$

*2.* $\frac{1}{\sqrt{m}} \left\| J(\mathbf{w})S \right\|_F \leq \kappa$

*Proof.* Using sub-multiplicativity of the Frobenius norm,

$$\frac{1}{\sqrt{m}} \left\| (J(\mathbf{w}) - J(\tilde{\mathbf{w}}))S \right\|_F$$

$$\leq \frac{1}{\sqrt{m}} \left\| (J(\mathbf{w}) - J(\tilde{\mathbf{w}})) \right\|_F \left\| S \right\|_F$$

$$\leq^{(1)} \left\| S \right\|_F \tilde{\kappa} \left\| \mathbf{w} - \tilde{\mathbf{w}} \right\| =: \kappa \left\| \mathbf{w} - \tilde{\mathbf{w}} \right\|$$

where $^{(1)}$ is given by Lemma (1) of Lee et al. (2019) and $\kappa := \tilde{\kappa} \left\| S \right\|_F$ which is constant since $\left\| S \right\|_F$ is bounded. The second part of the theorem can be shown using the same two arguments. □

**Lemma A.2.** *(Lemma 4.2 from the paper) For $\delta_0 > 0$, $\frac{2}{\lambda_{min}(KS) + \lambda_{max}(KS)} > \eta_0$ and $S$ such that $S \succ 0$, there exist $C > 0$, $N \in \mathbb{N}$ and $\kappa > 1$, such that for every $m \geq N$, the following holds with probability at least $1 - \delta_0$ over random initialization. When applying preconditioned gradient descent as in (7) with learning rate $\eta = \eta_0/m$*

*1.* $\left\| \mathbf{r}_t \right\|_2 \leq \left( 1 - \frac{\eta \lambda_{min}}{3} \right)^t C$

*2.* $\sum_{j=1}^{t} \left\| \mathbf{w}_j - \mathbf{w}_{j-1} \right\|_2 \leq \frac{3\kappa C}{\lambda_{min}} m^{-1/2}$

*3.* $\sup_t \left\| (K_0 - K_t)S \right\|_F \leq \frac{6\kappa^3 C}{\lambda_{min}} m^{-1/2}$,

*where $\lambda_{min}$ is the minimal eigenvalue of $KS$.*

*Proof.* We begin by proving parts (1) and (2) in the theorem with $\kappa$ defined by Lemma A.1. We do that by induction where the base case for $t = 0$ is clear. For $t > 0$ we have that

$$\left\| \mathbf{w}_{t+1} - \mathbf{w}_t \right\| \leq \eta \left\| J(\mathbf{w}_t)S \right\|_{op} \left\| \mathbf{r}_t \right\|_2 \leq \left( 1 - \frac{\eta_0 \lambda_{min}}{3} \right)^t R_0 \kappa \eta_0 m^{-1/2}$$

Implying that

$$\sum_{j=0}^{t-1} \left\| \mathbf{w}_{j+1} - \mathbf{w}_j \right\| \leq R_0 \kappa \eta_0 m^{-1/2} \sum_{j=0}^{t-1} \left( 1 - \frac{\eta_0 \lambda_{min}}{3} \right)^j \leq \frac{3\kappa R_0}{\lambda_{min}} m^{-1/2}$$

Which concludes part (2) of the theorem.

For part (1) we use the mean value theorem to get that

$$
\begin{aligned}
\|\mathbf{r}_{t+1}\| &= \|\mathbf{r}_{t+1} - \mathbf{r}_t + \mathbf{r}_t\| \\
&= \|J(\tilde{\mathbf{w}}_t)(\mathbf{w}_{t+1} - \mathbf{w}_t) + \mathbf{r}_t\| = \left\|\eta J(\tilde{\mathbf{w}}_t)J(\mathbf{w}_t)^T S \mathbf{r}_t + \mathbf{r}_t\right\| \\
&\leq \left\|I - \eta J(\tilde{\mathbf{w}}_t)J(\mathbf{w}_t)^T S\right\|_2 \|\mathbf{r}_t\| \\
&\leq \left\|I - \eta J(\tilde{\mathbf{w}}_t)J(\mathbf{w}_t)^T S\right\|_2 \left(1 - \frac{\eta_0 \lambda_{min}}{3}\right)^t R_0
\end{aligned}
$$

where $\tilde{\mathbf{w}}_t$ is some point on the line between $\mathbf{w}_{t+1}$ and $\mathbf{w}_t$. It remains to show that with probability at least $(1 - \delta_0/2)$ it holds that

$$
\left\|I - \eta J(\tilde{\mathbf{w}}_t)J(\mathbf{w}_t)^T S\right\|_2 \leq \left(1 - \frac{\eta_0 \lambda_{min}}{3}\right)
$$

We show that by observing that $S$ doesn't change the convergence in probability so $K_0 S \to K S$ as in Yang (2019), therefore we can choose large enough $m$ such that

$$
\|KS - K_0 S\|_F \leq \frac{\eta_0 \lambda_{min}}{3}
$$

Therefore we get

$$
\begin{aligned}
&\left\|I - \eta J(\tilde{\mathbf{w}}_t)J(\mathbf{w}_t)^T S\right\|_2 \\
&\leq \|I - \eta_0 K S\|_2 + \eta_0 \|(K - K_0)S\|_2 + \eta \left\|(J(\mathbf{w}_0)J(\mathbf{w}_0)^T - J(\tilde{\mathbf{w}}_t)J(\mathbf{w}_t))S\right\|_2 \\
&\leq (1 - \eta_0 \lambda_{min} + \frac{\eta_0 \lambda_{min}}{3} + 2\eta_0 \kappa^2 \frac{3\kappa R_0}{\lambda_{min}} m^{-1/2} \leq 1 - \frac{\eta_0 \lambda_{min}}{3}
\end{aligned}
$$

which concludes part (1).

For part (3) we verify that using Lemma A.1

$$
\|(K_0 - K_t)S\|_F = \frac{1}{m} \left\|(J(\mathbf{w}_0)J(\mathbf{w}_0)^T - J(\mathbf{w}_t)J(\mathbf{w}_t)^T)S\right\|_F \tag{20}
$$

$$
\leq \frac{1}{m} \left(\|J(\mathbf{w}_0)S\|_2 \left\|J(\mathbf{w}_0)^T - J(\mathbf{w}_t)^T\right\|_F + \|J(\mathbf{w}_t) - J(\mathbf{w}_0)\|_2 \left\|J(\mathbf{w}_t)^T S\right\|_F\right) \tag{21}
$$

$$
\leq 2\kappa^2 \|\mathbf{w}_0 - \mathbf{w}_t\|_2 \leq \frac{6\kappa^3 R_0}{\lambda_{min}} m^{-1/2} \tag{22}
$$

$\square$

Next we prove the preconditioned dynamics given in 4.1

**Theorem A.3.** *(Theorem 4.1 from the paper) Let $\delta_0 > 0$, $\frac{2}{\lambda_{min}(KS) + \lambda_{max}(KS)} > \eta_0$, $\epsilon > 0$ and $S$ such that $S \succ 0$. Then, there exists $N \in \mathbb{N}$ such that for every $m \geq N$, the following holds with probability at least $1 - \delta_0$ over random initialization when applying preconditioned gradient descent with learning rate $\eta = \eta_0/m$*

$$
\mathbf{r}_t = (I - \eta_0 K S)^t \mathbf{y} \pm \xi(t),
$$

*where $\|\xi\|_2 \leq \epsilon$.*

*Proof.* By using the mean value theorem we have that

$$
\begin{aligned}
\mathbf{r}_{t+1} &= \mathbf{r}_{t+1} + \mathbf{r}_t - \mathbf{r}_t = J(\tilde{\mathbf{w}})(\mathbf{w}_{t+1} - \mathbf{w}_t) + \mathbf{r}_t \\
&=^{(1)} -\eta J(\tilde{\mathbf{w}})J(\mathbf{w}_t)^T S \mathbf{r}_t + \mathbf{r}_t = (I - \eta J(\mathbf{w}_t)J(\mathbf{w}_t)^T S)\mathbf{r}_t - \eta(J(\tilde{\mathbf{w}}) - J(\mathbf{w}_t))J(\mathbf{w}_t)^T S \mathbf{r}_t \\
&=^{(2)} (I - \eta_0 K S)\mathbf{r}_t - \underbrace{\eta_0((K_t - K)S)\mathbf{r}_t - \eta(J(\tilde{\mathbf{w}}) - J(\mathbf{w}_t))J(\mathbf{w}_t)^T S \mathbf{r}_t}_{\epsilon(t)}
\end{aligned}
$$

where (1) is by the gradient descent definition and (2) is by $K_t$ definition.

Now applying the theorem inductively we get

$$\mathbf{r}_{t+1} = (I - \eta_0 KS)^t \mathbf{r}_0 + \underbrace{\sum_{i=1}^{t} (I - \eta_0 KS)^i \epsilon(t-i)}_{\xi(t)}$$

Next we bound the norm of $\epsilon(t) = \underbrace{\eta_0(K_t - K)S\mathbf{r}_t}_{A} - \underbrace{\eta(J(\tilde{\mathbf{w}}) - J(\mathbf{w}_t))J(\mathbf{w}_t)^T S\mathbf{r}_t}_{B}$. For term $A$ we have

$$\|\eta_0(K_t - K)S\mathbf{r}_t\|_2 \leq \eta_0(\|(K_t - K_0)S\|_{op} + \|(K_0 - K)S\|_{op})\|\mathbf{r}_t\|_2$$

$$\leq^{(1)} \eta_0(\frac{6\kappa^3 R_0}{\lambda_{min}} m^{-1/2} + \frac{\lambda_{min}\epsilon}{2R_0})R_0 = \frac{6\kappa^3 R_0}{\lambda_{min}} m^{-1/2} + \frac{\epsilon\lambda_{min}\eta_0}{2}$$

where (1) follows from Lemma 4.2 and the fact that $K_0 S \to KS$ in probability Yang (2019). So (1) holds when $m$ is sufficiently large.

For term $B$ we have that

$$\left\|\eta(J(\tilde{\mathbf{w}}) - J(\mathbf{w}_t))J(\mathbf{w}_t)^T S\mathbf{r}_t\right\| \leq \frac{\eta_0}{m}\|J(\tilde{\mathbf{w}}) - J(\mathbf{w}_t)\|_{op}\|J(\mathbf{w}_t)S\mathbf{r}_t\|_2$$

$$\leq \frac{\eta_0}{m}\|J(\tilde{\mathbf{w}}) - J(\mathbf{w}_t)\|_{op}\|J(\mathbf{w}_t)S\|_{op}\|\mathbf{r}_t\|_2 \leq^{(1)} \eta_0 \kappa^2 \|\tilde{\mathbf{w}} - \mathbf{w}_t\| R_0$$

$$\leq^{(2)} \eta_0 \kappa \|\mathbf{w}_{t+1} - \mathbf{w}_t\| R_0 \leq \frac{\eta_0^2 \kappa^3 R_0^3}{\sqrt{m}} \tag{23}$$

where (1) is by Lemma A.1 and (2) it by Lemma 4.2. Therefore we have

$$\mathbf{r}_t = (I - \eta KS)^t \mathbf{y} + \xi(t)$$

where

$$\|\xi(t)\| = \left\|\sum_{i=1}^{t}(I - \eta_0 KS)^i \epsilon(t-i) + (I - \eta KS)^t f(X, \mathbf{w}_0)\right\|$$

$$\leq \left\|\sum_{i=1}^{t}(I - \eta_0 KS)^i\right\|_{op} (\frac{\eta_0^2 \kappa^3 R_0^3}{\sqrt{m}} + \frac{6\kappa^3 R_0}{\lambda_{min}\sqrt{m}} + \frac{\epsilon\lambda_{min}\eta_0}{2}) + \left\|(I - \eta KS)^t f(X, \mathbf{w}_0)\right\|$$

$$\leq (\frac{\eta_0^2 \kappa^3 R_0^3}{\sqrt{m}} + \frac{6\kappa^3 R_0}{\lambda_{min}\sqrt{m}} + \frac{\epsilon\lambda_{min}\eta_0}{2})\sum_{i=1}^{t}(1 - \eta_0\lambda_{min})^i + (1 - \eta_0\lambda_{min})^t \nu C_0$$

$$\leq (\frac{\eta_0^2 \kappa^3 R_0^3}{\sqrt{m}} + \frac{6\kappa^3 R_0}{\lambda_{min}\sqrt{m}} + \frac{\epsilon\lambda_{min}\eta_0}{2})\frac{1}{\eta_0\lambda_{min}} + (1 - \eta_0\lambda_{min})^t \nu C_0$$

$$= \epsilon/2 + \frac{\eta_0^2 \kappa^3 R_0^3}{\sqrt{m}} + \frac{6\kappa^3 R_0}{\lambda_{min}\sqrt{m}} + (1 - \eta_0\lambda_{min})^t \nu C_0$$

So by choosing $m$ to be large enough and $\nu < \frac{\epsilon}{3C_0}$ we conclude the theorem. $\qquad\square$

## B  Convergence Analysis

Under the settings of Sec. 4.2, we here prove the following lemma.

**Lemma B.1.** *(Lemma 4.3 from the paper)*

> *1. Let $\mathbf{w}_0 \in \mathbb{R}^p$ and $\phi(\mathbf{x}) := \nabla f(\mathbf{x}, \mathbf{w}_0)$, it holds that $\mathbf{w}^* = \mathbf{w}^{**}$.*

2. *Let $\delta_0 > 0$, $\eta_0 < \frac{2}{\lambda_{min}(KS) + \lambda_{max}(KS)}$ , $\epsilon > 0$, $x \in \mathbb{R}^d$ be a test point, $T > 0$ number of iterations and $S$ such that $S \succ 0$. Then there exists $N \in \mathbb{N}$ such that for every $m \geq N$, the following holds with probability at least $1 - \delta_0$ over random initialization of $\mathbf{w}_0$ when applying preconditioned gradient descent to both $f$ and $h$ with learning rate $\eta = \eta_0/m$*

$$|h(\mathbf{x}, \mathbf{w}'_T) - f(\mathbf{x}, \mathbf{w}_T)| \leq \epsilon,$$

*where $\mathbf{w}'$ is as in* (13).

*Proof.* For point (1), applying lemma B.2 with the kernel $\boldsymbol{k}(\mathbf{x}, \mathbf{z}) = \langle \phi(\mathbf{x}), \phi(\mathbf{x}) \rangle$ (and kernel matrix $K_0$) yields $h(\cdot, w^*) = \sum_{i=1}^n \alpha_i^* \langle \phi(\mathbf{x}), \phi(\mathbf{x}_i) \rangle$ with

$$\alpha^* = \left(K_0 + \gamma S^{-1}\right)^{-1} \mathbf{y},$$

which uniquely determines that $w^* = \sum_{i=1}^n \alpha_i^* \phi(\mathbf{x}_i)$.

When $\gamma \to 0$ this becomes the same minimizer as of the standard kernel ridge regression problem with $L(\mathbf{w}) = \frac{1}{2} \|f(X, \mathbf{w}) - \mathbf{y}\|_2^2$ (or equivalently, substituting $S = I$).

Point (2) is proved in lemma B.3. $\qquad\square$

**Lemma B.2.** *Let $\boldsymbol{k}(\mathbf{x}, \mathbf{z}) = \langle \phi(\mathbf{x}), \phi(\mathbf{z}) \rangle$ be any kernel and $[K]_{ij} = \frac{1}{n} \boldsymbol{k}(\mathbf{x}_i, \mathbf{x}_j)$ be its kernel matrix. Given $S \succ 0$, let $L_S(f) = \frac{1}{2} \left\| S^{\frac{1}{2}}(f(X) - \mathbf{y}) \right\|_2^2 + \frac{\gamma}{2} \|f\|_{\mathcal{H}}^2$ (where $\gamma > 0$ is some regularization term). Then the unique minimizer $f^* = \arg\min_{\mathbf{w}} L(\mathbf{w})$ takes the from $f^*(\mathbf{x}) = \sum_{i=1}^n \alpha_i^* \langle \phi(\mathbf{x}), \phi(\mathbf{x}_i) \rangle$ with*

$$\alpha^* = \left(K + \gamma S^{-1}\right)^{-1} \mathbf{y}$$

*Proof.* The Representer Theorem (Schölkopf et al., 2001) states that any minimizer of the kernel ridge regression problem $f^* = \arg\min_{\mathbf{w}} L_S(f)$ is of the form $f^*(\mathbf{x}) = \sum_{i=1}^n \alpha_i^* \langle \phi(\mathbf{x}), \phi(\mathbf{x}_i) \rangle$.

Then $f^*(X)$ can be written as $K\alpha^*$ and $\|f^*\|_{\mathcal{H}}^2 = \alpha^{*T} K \alpha^*$. So we can equivalently solve for:

$$\alpha^* = \arg\min_{\alpha \in \mathbb{R}^n} \frac{1}{2} \left\| S^{\frac{1}{2}}(K\alpha - \mathbf{y}) \right\|_2^2 + \frac{\gamma}{2} \alpha^T K \alpha,$$

where $K_{ij} = \langle (\phi(\mathbf{x}_i), \phi(\mathbf{x}_j) \rangle$. Now opening up the norm we get:

$$\alpha^* = \arg\min_\alpha \frac{1}{2} \left(\alpha^T K^T S K \alpha - \alpha^T K^T S \mathbf{y} - \mathbf{y}^T S K \alpha + \mathbf{y}^T S \mathbf{y}\right) + \frac{\gamma}{2} \alpha^T K \alpha.$$

Then as $K, S$ are symmetric, and $\mathbf{y}^T S \mathbf{y}$ doesn't depend on $\alpha$ the above simplifies to:

$$\alpha^* = \arg\min_\alpha \frac{1}{2} \alpha^T K^T S K \alpha - \alpha^T K^T S \mathbf{y} + \frac{\gamma}{2} \alpha^T K \alpha$$

$$= \arg\min_\alpha \frac{1}{2} \alpha^T \left(KSK + \gamma K\right) \alpha - \alpha^T K S \mathbf{y}.$$

Now deriving with respect to $\alpha$ and setting to 0 we obtain:

$$0 = (KSK + \gamma K) \alpha - KS\mathbf{y} = KS \left(\left(K + \gamma S^{-1}\right) \alpha - \mathbf{y}\right).$$

As $K \succ 0$ and $S \succ 0$ it must hold that:

$$0 = \left(K + \gamma S^{-1}\right) \alpha - \mathbf{y}.$$

Since the loss is convex, $\alpha^* = \left(K + \gamma S^{-1}\right)^{-1} \mathbf{y}$ is the unique minimizer. $\qquad\square$

**Lemma B.3.** *For $\delta_0 > 0$, $\frac{2}{\lambda_{min}(KS)+\lambda_{max}(KS)} > \eta_0$ , $\epsilon > 0$, $x \in \mathbb{R}^d$ be a test point, $T > 0$ number of iterations and $S$ such that $S \succ 0$, there exists $N \in \mathbb{N}$ such that for every $m \geq N$, the following holds with probability at least $1 - \delta_0$ over random initialization of $\mathbf{w}_0$ when applying preconditioned gradient descent to both $f$ and $h$ with learning rate $\eta = \eta_0/m$*

$$|\langle \mathbf{w}'_T, \nabla f(\mathbf{x}, \mathbf{w}_0) \rangle - f(\mathbf{x}, \mathbf{w}_T)| < \epsilon$$

*where $\mathbf{w}'$ is as in* (13).

*Proof.* Let $\mathbf{x}, y$ be some data point and its label, which are not necessarily in the train set. Denote by $\tilde{\mathbf{r}}_t = f(\mathbf{x}, \mathbf{w}_t) - y$, and $\tilde{J}(\mathbf{w}) = \nabla f(\mathbf{x}, \mathbf{w})$. We define a linearized model $f^{lin}(\mathbf{x}, \mathbf{w}'_t) = f(\mathbf{x}, \mathbf{w}_0) + \tilde{J}(\mathbf{w}_0)\mathbf{w}'_t$ where $\mathbf{w}'_t$ is as in (13). In particular, note that $f^{lin}(\mathbf{x}, \mathbf{w}'_0) = f(\mathbf{x}, \mathbf{w}_0)$.

Let $h(\mathbf{x}, \mathbf{w}) = \langle \mathbf{w}, \nabla f(\mathbf{x}, \mathbf{w}_0) \rangle$. By the triangle inequality,

$$|h(\mathbf{x}, \mathbf{w}'_T) - f(\mathbf{x}, \mathbf{w}_T)| \leq |f^{lin}(\mathbf{x}, \mathbf{w}'_T) - f(\mathbf{x}, \mathbf{w}_T)| + |f(\mathbf{x}, \mathbf{w}_0)|$$

The weights of the last layer are initialized $\mathcal{N}(0, \nu)$ with some constant $\nu$ which we may choose. Thus, by taking small enough $\nu$, $|f(\mathbf{x}, \mathbf{w}_0)| < \frac{\epsilon}{3}$ with the desired probability. Notice that $\tilde{\mathbf{r}}_t^{lin} - \tilde{\mathbf{r}}_t = f^{lin}(\mathbf{x}, \mathbf{w}'_t) - f(\mathbf{x}, \mathbf{w}_t)$ and as such, it remains to show that $|\tilde{\mathbf{r}}_T^{lin} - \tilde{\mathbf{r}}_T| < \frac{2\epsilon}{3}$.

By the mean value theorem, there exists some $\tilde{\mathbf{w}}$ s.t

$$
\begin{aligned}
\tilde{\mathbf{r}}_{t+1} - \tilde{\mathbf{r}}_t &= \tilde{J}(\tilde{\mathbf{w}})(\mathbf{w}_{t+1} - \mathbf{w}_t) \\
&=^{(1)} -\eta \tilde{J}(\tilde{\mathbf{w}})J(\mathbf{w}_t)^T S\mathbf{r}_t = -\eta \tilde{J}(\mathbf{w}_t)J(\mathbf{w}_t)^T S\mathbf{r}_t - \eta(\tilde{J}(\tilde{\mathbf{w}}) - \tilde{J}(\mathbf{w}_t))J(\mathbf{w}_t)^T S\mathbf{r}_t \\
&=^{(2)} -\eta_0 K_0(\mathbf{x}, X)S\mathbf{r}_t - \underbrace{\eta_0((K_t(\mathbf{x}, X) - K_0(\mathbf{x}, X))S)\mathbf{r}_t - \eta(\tilde{J}(\tilde{\mathbf{w}}) - \tilde{J}(\mathbf{w}_t))J(\mathbf{w}_t)^T S\mathbf{r}_t}_{\epsilon(\mathbf{x},t)}
\end{aligned}
$$

where (1) is by the gradient descent definition and (2) is by $K_t$ definition.

Thus, taking a telescopic series we get that for every $t$,

$$\tilde{\mathbf{r}}_T = \tilde{\mathbf{r}}_0 + \sum_{i=0}^{T-1} \tilde{\mathbf{r}}_{i+1} - \tilde{\mathbf{r}}_i = \tilde{\mathbf{r}}_0 - \eta_0 \sum_{i=0}^{T-1} (K_0(\mathbf{x}, X)S\mathbf{r}_i + \epsilon(\mathbf{x}, i))$$

Now for the linear model, by definition of our PGD updates we have

$$\mathbf{w}'_{t+1} - \mathbf{w}'_t = -\eta J(\mathbf{w}'_t)^T S\mathbf{r}_t^{lin} = -\eta J(\mathbf{w}_0)^T S\mathbf{r}_t^{lin}$$

and by definition of the linearized model,

$$
\begin{aligned}
f^{lin}(\mathbf{x}, \mathbf{w}'_{t+1}) - f^{lin}(\mathbf{x}, \mathbf{w}'_t) &= f(\mathbf{x}, \mathbf{w}_0) + \tilde{J}(\mathbf{w}_0)(\mathbf{w}'_{t+1}) - (f(\mathbf{x}, \mathbf{w}_0) + \tilde{J}(\mathbf{w}_0)\mathbf{w}'_t) \\
&= \tilde{J}(\mathbf{w}_0)(\mathbf{w}'_{t+1} - \mathbf{w}'_t) = -\eta \tilde{J}(\mathbf{w}_0)J(\mathbf{w}_0)^T S\mathbf{r}_t^{lin} \\
&= -\eta_0 K_0(\mathbf{x}, X)S\mathbf{r}_t^{lin}
\end{aligned}
$$

And so

$$\tilde{\mathbf{r}}_{t+1}^{lin} - \tilde{\mathbf{r}}_t^{lin} = -\eta_0 K_0(\mathbf{x}, X)S\mathbf{r}_t^{lin}$$

Thus, taking a telescopic series we get that for every $t$,

$$\tilde{\mathbf{r}}_T^{lin} = \tilde{\mathbf{r}}_0^{lin} + \sum_{i=0}^{T-1} \tilde{\mathbf{r}}_{i+1}^{lin} - \tilde{\mathbf{r}}_i^{lin} = \tilde{\mathbf{r}}_0^{lin} - \eta_0 \sum_{i=0}^{T-1} K_0(\mathbf{x}, X)S\mathbf{r}_i^{lin}$$

Now we can compare between the linearized and non linearized versions. Since $\tilde{\mathbf{r}}_0^{lin} = \tilde{\mathbf{r}}_0$ we have

$$\tilde{\mathbf{r}}_T^{lin} - \tilde{\mathbf{r}}_t = -\eta_0 K_0(\mathbf{x}, X)S\left(\sum_{i=0}^{T-1} \mathbf{r}_i^{lin} - \mathbf{r}_i\right) + \left(\sum_{i=0}^{T-1} \epsilon(\mathbf{x}, i)\right)$$

Notice that $\mathbf{r}_i^{lin} = (I - \eta_0 K_0 S)^i \mathbf{r}_0$. Theorem 4.1 combined with choosing $\nu$ sufficiently small, states that $\mathbf{r}_i = (I - \eta_0 K S)^i \mathbf{r}_0 \pm \xi(i)$ with $\|\xi_i\| \xrightarrow[m\to\infty]{} 0$. Analogously, one could also show $\mathbf{r}_i = (I - \eta_0 K_0 S)^i \mathbf{r}_0 \pm \xi(i)$ by replacing $K$ in the proof with $K_0$. As such, for every $i < T$, $\left\| \mathbf{r}_i^{lin} - \mathbf{r}_i \right\| \leq \|\xi(i)\|$ with $\|\xi_i\| \xrightarrow[m\to\infty]{} 0$. Now by Cauchey Shwartz and the triangle inequality we get:

$$\left| \tilde{\mathbf{r}}_T^{lin} - \tilde{\mathbf{r}}_T \right| \leq \|\eta_0 K_0(\mathbf{x}, X) S\| \left( \sum_{i=0}^{T-1} \left\| \mathbf{r}_i^{lin} - \mathbf{r}_i \right\| \right) + \left| \sum_{i=0}^{T-1} \epsilon(\mathbf{x}, i) \right|$$

$$= \|\eta_0 K_0(\mathbf{x}, X) S\| \left( \sum_{i=0}^{T-1} \|\xi(i)\| \right) + \left| \sum_{i=0}^{T-1} \epsilon(\mathbf{x}, i) \right|$$

$$\leq \|\eta_0 K_0(\mathbf{x}, X) S\| \, T \max_{0 \leq i \leq T-1} \|\xi(i)\| + T \left| \max_{0 \leq i \leq T-1} \epsilon(\mathbf{x}, i) \right|.$$

Now each $\xi(i)$ tends to 0 as $m$ tends to infinity. So there exists some $N_0 \in \mathbb{N}$ s.t for all $m > N_0$, $\max_{0 \leq i \leq T-1} \|\xi(i)\| < \frac{\epsilon}{3T\|\eta_0 K_0(\mathbf{x},X)S\|}$ so that the entire left hand side is at most $\frac{\epsilon}{3}$.

It remains to bound $|\max_{0 \leq i \leq T-1} \epsilon(\mathbf{x}, i)|$. Fix some $t$ to be the argmax. Recalling the definition of $\epsilon(\mathbf{x}, t)$ and using the fact that $\mathbf{r}_t$ is bounded by Lemma 4.2, it suffices to bound $(K_0(\mathbf{x}, X) - K_t(\mathbf{x}, X)S)$ and $(\tilde{J}(\tilde{\mathbf{w}}) - \tilde{J}(\mathbf{w}_t))J(\mathbf{w}_t)^T)S$.

$$\|(K_0(\mathbf{x}, X) - K_t(\mathbf{x}, X)S\|_F = \frac{1}{m} \left\| (\tilde{J}(\mathbf{w}_0)J(\mathbf{w}_0)^T - J(x, \mathbf{w}_t)J(\mathbf{w}_t)^T)S \right\|_F$$

$$= \frac{1}{m} \left\| (\tilde{J}(\mathbf{w}_0)J(\mathbf{w}_0)^T - \tilde{J}(\mathbf{w}_0)J(\mathbf{w}_t)^T + (\tilde{J}(\mathbf{w}_0)J(\mathbf{w}_t)^T - \tilde{J}(\mathbf{w}_t)J(\mathbf{w}_t)^T)S \right\|_F$$

$$\leq \frac{1}{m} \left( \left\| \tilde{J}(\mathbf{w}_0) \right\|_2 \left\| (J(\mathbf{w}_0)^T - J(\mathbf{w}_t)^T)S \right\|_F + \left\| \tilde{J}(\mathbf{w}_0) - \tilde{J}(\mathbf{w}_t) \right\|_2 \left\| J(\mathbf{w}_t)^T S \right\|_F \right)$$

$$\leq 2\kappa^2 \left\| \mathbf{w}_0 - \mathbf{w}_t \right\|_2 \leq \frac{6\kappa^3 R_0}{\lambda_{min}} m^{-1/2},$$

where the last inequality holds since Lemma A.1 holds analogously for $\tilde{J}$ and using Lemma 4.2 for $\|\mathbf{w}_0 - \mathbf{w}_t\|_2$. $(\tilde{J}(\tilde{\mathbf{w}}) - \tilde{J}(\mathbf{w}_t))J(\mathbf{w}_t)^T)S$ can be similarly be bound by applying by Lemma A.1 analogously to (23). As such there is some $N_1$ s.t the entire right hand side is at most $\frac{\epsilon}{3}$.

Taking $N = \max\{N_0, N_1\}$ completes the proof. $\qquad\square$

## C Consistency of Spectral Engineering

**Theorem C.1.** *(Theorem 3.2 from the paper) Let $\mathbf{k}(\mathbf{x}, \mathbf{z}) = \sum_{k=1}^{\infty} \lambda_k \Phi_k(\mathbf{x}) \Phi_k(\mathbf{z})$ and $\mathbf{k}_g(\mathbf{x}, \mathbf{z}) = \sum_{k=1}^{\infty} g(\lambda_k) \Phi_k(\mathbf{x}) \Phi_k(\mathbf{z})$ be two Mercer kernels with non-zero eigenvalues $\{\lambda_i\}, \{g(\lambda_i)\}$ and eigenfunctions $\{\Phi_i\}$ such that $\forall \mathbf{x} \in \mathcal{X}, |\Phi_i(\mathbf{x})| \leq M$. Assuming $g$ is $L$ Lipchitz. Let $K, K_g$ be the corresponding kernel matrices on i.i.d samples $\mathbf{x}_1, .., \mathbf{x}_n \in \mathcal{X}$. Define the kernel matrix $\tilde{K}_g = V D V^T$ where $V = (\mathbf{v}_1, .., \mathbf{v}_n)$ with $\mathbf{v}_i$ the i'th eigenvector of $K$ and $D$ is a diagonal matrix with $D_{ii} = g(\hat{\lambda}_i)$ where $\hat{\lambda}_i$ is the i'th eigenvalue of $K$. Then, for $n \to \infty$*

$$\left\| \tilde{K}_g - K_g \right\|_F \xrightarrow{a.s} 0,$$

*where a.s. stands for almost surely.*

*Proof.* From the Mercer decomposition of $\mathbf{k}_g$ we have that

$$K_g = \sum_{k=1}^{\infty} g(\lambda_k) \Phi_k(X) \Phi_k^T(X)$$

where $\Phi_k(X) = \frac{1}{\sqrt{n}}(\Phi_k(x_1),..,\Phi_k(x_n))^T \in \mathbb{R}^n$. We also define

$$K_g^{\leq R} := \sum_{k=1}^{R} g(\lambda_k) \Phi_k(X) \Phi_k^T(X)$$

$$K_g^{>R} := \sum_{k=R}^{\infty} g(\lambda_k) \Phi_k(X) \Phi_k^T(X)$$

Let $\mathbf{v}_1,..,\mathbf{v}_n$ be the eigenvectors of $K$ (and therefore of $\tilde{K}_g$) and fix $R > 0$. Then we have

$$\left\|\tilde{K}_g - K_g\right\|_F \leq \left\|\tilde{K}_g - K_g^{\leq R}\right\|_F + \left\|K_g^{>R}\right\|_F = \left\|K_g^{>R}\right\|_F + \sqrt{\sum_{j=1}^{n} \langle (\tilde{K}_g - K_g^{\leq R})\mathbf{v}_j, (\tilde{K}_g - K_g^{\leq R})\mathbf{v}_j \rangle}$$

$$= \left\|K_g^{>R}\right\|_F + \sqrt{\sum_{j=1}^{n} \langle g(\hat{\lambda}_j)\mathbf{v}_j - K_g^{\leq R}\mathbf{v}_j, g(\hat{\lambda}_j)\mathbf{v}_j - K_g^{\leq R}\mathbf{v}_j \rangle}$$

$$= \left\|K_g^{>R}\right\|_F + \sqrt{\sum_{j=1}^{n} \left( g(\hat{\lambda}_j)^2 - 2g(\hat{\lambda}_j)\langle \mathbf{v}_j, K_g^{\leq R}\mathbf{v}_j \rangle + \langle K_g^{\leq R}\mathbf{v}_j, K_g^{\leq R}\mathbf{v}_j \rangle \right)}$$

$$= \left\|K_g^{>R}\right\|_F$$

$$+ \left( \sum_{j=1}^{n} \left( g(\hat{\lambda}_j)^2 - 2g(\hat{\lambda}_j) \sum_{k=1}^{R} g(\lambda_k)(\mathbf{v}_j^T \Phi_k(X))^2 + \sum_{k,l=1}^{R} g(\lambda_k)g(\lambda_l)(\mathbf{v}_j^T\Phi_k(X))(\mathbf{v}_j^T\Phi_l(X))\Phi_l(X)^T\Phi_k(X) \right) \right)^{\frac{1}{2}}$$

Since $\Phi_1, \Phi_2,...$ are orthonormal, and $\Phi_i(\mathbf{x}_1),..,\Phi_i(\mathbf{x}_n)$ are i.i.d, by the law of large numbers it holds that $\Phi_l(X)^T\Phi_k(X) \to \mathbb{E}\Phi_l\Phi_k = \delta_{ij}$, therefore, the last term converges a.s to

$$\left\|K_g^{>R}\right\|_F + \left( \sum_{j=1}^{n} \left( g(\hat{\lambda}_j)^2 - 2g(\hat{\lambda}_j) \sum_{k=1}^{R} g(\lambda_k)(\mathbf{v}_j^T \Phi_k(X))^2 + \sum_{k=1}^{R} g(\lambda_k)^2(\mathbf{v}_j^T\Phi_k(X))^2 \right) \right)^{\frac{1}{2}}$$

Next we use Braun (2005)[Theorem 4.10 and Equation 4.15] to get that

$$\sum_{i=1}^{d}(\mathbf{v}_i^T\Phi(X))^2 \to \begin{cases} 1 & \text{if } \Phi \text{ is an eigenfunction of } \lambda_k \\ 0 & \text{else} \end{cases}$$

Which implies that

$$\sum_{k=1}^{R} g(\lambda_k) \cdot (\mathbf{v}_j^T\Phi_k(X))^2 \to \delta_{j,k}g(\lambda_k)$$

. Therefore we get that

$$\left\|\tilde{K}_g - K_g\right\|_F \leq \left\|\tilde{K}_g - K_g^{\leq R}\right\|_F + \left\|K_g^{>R}\right\|_F \to \sqrt{\sum_{i=1}^{R}(g(\hat{\lambda}_j) - g(\lambda_j))^2 + \sum_{i=R}^{n} g(\hat{\lambda}_j)^2} + \left\|K_g^{>R}\right\|_F$$

Finally, from the Lipchitzness, $\sum_{i=1}^{R}(g(\hat{\lambda}_j) - g(\lambda_j))^2 \leq L^2 \sum_{i=1}^{R}(\hat{\lambda}_j - \lambda_j)^2$ so by applying Rosasco et al. (2010) (Proposition 10) we get

$$\sqrt{\sum_{i=1}^{R}(g(\hat{\lambda}_j) - g(\lambda_j))^2 + \sum_{i=R}^{n} g(\hat{\lambda}_j)^2} + \left\|K_g^{>R}\right\|_F \to \left\|K_g^{>R}\right\|_F + \sqrt{\sum_{i=R}^{n} g(\hat{\lambda}_j)^2}$$

Observing that $\left\|K_g^{>R}\right\|_F + \sqrt{\sum_{i=R}^{n} g(\hat{\lambda}_j)^2} \to_{R\to\infty} 0$ give us the desired result. $\square$

