# OpenReview forum: "Controlling the Inductive Bias of Wide Neural Networks by Modifying the Kernel’s Spectrum"
_TMLR — Accepted by TMLR_

### Review · Reviewer_wmuT · 2023-09-12

**Summary Of Contributions:**

The manuscript studies the problem of controlling the inductive bias of wide neural networks by modifying the NTK kernel’s spectrum. This is done through the introduction of Modified Spectrum Kernels (MSKs), a family of constructed kernels that can be used to approximate kernels with desired eigenvalues for which a closed form is unknown. The approach then proposes a preconditioned gradient descent method, which alters the trajectory of gradient descent, yielding significant convergence speedups for training wide neural networks without changing the final solution.

**Audience:**

Yes

**Claims And Evidence:**

Yes

**Requested Changes:**

- There is a duplicate sentence on Page 6
- [Optional] I suggest making the example (making the top k + 1 eigenvalues equal to each other) in Section 4.3 an indexed result (i.e., a Remark/Corrolary).

**Strengths And Weaknesses:**

### Strengths

- In general, the paper is well-written. Background information and relevant literature are adequately covered. Technical developments of the work are rigorous, intuitive, and well-founded from existing works.
- The paper introduces several major technical contributions to the problem of interest, including:
    - using MSKs to manipulate the spectrum of existing kernels
    - using preconditioned gradient descent method to alter the trajectory of gradient descent toward the modified kernels (and thus control the spectral bias of wide neural networks)

    The proposed approaches would be of broad interest in the machine-learning community from both theoretical and algorithmic viewpoints.

- The mathematical quality of the work is high. Proofs seem to be rigorous and correct.
- The manuscript also obtained several results with interesting implications:
    - The example in Section 4.3 (making the top k + 1 eigenvalues equal to each other) leads to an exponential improvement over vanilla gradient descent (in learning the same eigenfunction).
    - While the results of Theorem 4.1 seem to be of the same spirit as the main results of (Arora 2019), they utilize the framework of (Lee et al 2019) for deep networks and the coverage of the result is much broader.

### Weaknesses

- No weakness was noted. One minor point is that I feel the result of Section 4.3 (which would be one with the greatest practical implications) is quite underwhelming in comparison to prior developments. Little information/discussion is provided on the required properties and the effects of the control function $g$ beyond the given example.

---

> ### Author Response · Authors · 2023-12-05
> **Response to Reviewer wmuT**
>
> We thank the reviewer for their positive feedback.
>
> We have expanded the discussion in Section 4.3, and added another experiment comparing various natural choices of $g$ which were not previously discussed in the paper. It includes letting $g(\lambda)=\lambda+c$ for some constant $c>0$, implying that $KS=K+cI$. Thus, we obtain a direct correspondence between PGD with such a choice of $g$, and \emph{regularized} kernel regression. Such a choice of $g$ may help prevent overfitting, even without early stopping. We also let $g$ to be constant, or $g(\lambda)=\sqrt{\lambda}$. Interestingly, it is observed that slower eigenvalue decays help prevent overfitting noise, as suggested by [R1, R2]. The results are summarized in Figure 2 and in the text. We elaborated the discussion on $g$.
>
> Thank you for spotting the typo on page 6. It has been corrected.
>
> We appreciate your suggestion regarding section 4.3 and adopted it, with the mentioned result now given by a Corollary.
>
> [R1] - Mallinar et al. 2022, Benign, Tempered, or Catastrophic: A Taxonomy of Overfitting
>
> [R2] - Bartlett et al. 2019, Benign Overfitting in Linear Regression

---

> > ### Comment · Reviewer_wmuT · 2023-12-06
> > **Figure 2**
> >
> > I want to thank the authors for the great revision of Section 4.3, which addressed all of my comments from the previous review.
> >
> > I have only one other comment: I think the experiments which lead to Figure 2 are great. However, the figure is uninformative about the green and orange curves. Perhaps it would be helpful to present the y-axis in a log scale instead? (I suppose that the testing MSE is close to zero but not equal to zero.)

---

> > > ### Author Response · Authors · 2023-12-06
> > > **Figure 2**
> > >
> > > We tried plotting the y axis in log scale. You can see how it looks on this anonymous GitHub we opened https://github.com/anon-git4462/Paper1479/blob/main/benign_overfitting_log_y.png
> > >
> > > It does distinguish between the green and orange curves, but it makes the red and blue curves look very similar. Each figure has an advantage and disadvantage. Perhaps we can leave the figure in the paper as it is, and add in text the values that the orange and green plots reach? We would appreciate your opinion.

---

> > > > ### Comment · Reviewer_wmuT · 2023-12-06
> > > >
> > > > Thanks for the figure! The seperation in scales between the orange and green curves is indeed unexpected.
> > > >
> > > > I find it very strange that the green curve already started with machine error (a MSE of 1e-30 over 1000x25 data point means each of the prediction is at machine error) with a very low number of samples (5-10 samples). Could you clarify a bit on this effect? If we take a step back and think about the setting: We fit 10 training samples with noisy Gaussian data and the testing error is exactly 0 on all the test data, it usually means that the method we used is bad.

---

> > > > > ### Author Response · Authors · 2023-12-07
> > > > > **Figure 2**
> > > > >
> > > > > Thank you for this comment. We realize that the case $g(\lambda)=1$ is in fact not interesting since it approximates the delta kernel which generalizes badly whether the training points are noisy or not. The plot for this choice is misleading -- the target function used is the zero function, which is always predicted by the delta kernel for all points that are not included in the training set. Therefore the prediction error is practically zero for any number of training points. We have therefore removed the plot for the $g=1$ function and left the other three selections of $g$ in.

---

> > > > > > ### Comment · Reviewer_wmuT · 2023-12-07
> > > > > >
> > > > > > That makes sense. Thanks again for your quick responses. I have no other questions or concerns.

---

### Review · Reviewer_VKSy · 2023-11-13

**Summary Of Contributions:**

This paper considered a preconditioned gradient descent (PGD) for wide neural networks (NNS) to accelerate the speed of convergence rate of PD in each eigenspace of the NTK. The authors proved the convergence rate of this PGD and showed that the predictor of GD training is approximately equivalent to the PGD training when the width is sufficiently large. Inspired by the spectral bias of NNs, the authors provided a new algorithm for PGD to control the spectral bias and gain a faster convergence rate for NNs.

**Audience:**

Yes

**Broader Impact Concerns:**

Not very applicable here.

**Claims And Evidence:**

Yes

**Requested Changes:**

1. What would be the dataset assumptions in Theorem 4.1 and Lemma 4.2? Can you go beyond compactly supported datasets and consider datasets like isotropic Gaussian datasets? What is the random initialization of NN in Theorem 4.1 and Lemma 4.2? There should be some discussion in the Section 4.1.
2. In Theorem 3.2, $(3.1)$ should be change to Definition 3.1.
3. On Page 3 above Kernel Construction with MSKs, the equation related to $\sum_{i=l}^d(v_i^\top \Phi(X))^2$ should be double-checked. What is $l$ here?
4. On Page 6, when discussing the second-order method with the Fisher information matrix, why can we consider $S=K_t^{-1}$? Does that mean we can consider preconditioning $S$ changing over the time $t$? This requires more discussion. Why PGD method in this paper can cover this method?
5. The second paragraph on Page 6 is repeated with the first paragraph. Please rewrite this part.
6. In the proof of Lemma A.1, how do you ensure $\\|S\\|_F$ is bounded which is not clear to me? Please provide more details for this proof or if there is some assumption on $S$.
7. What is $\bar \lambda_{\min}$ in the proof of Lemma A.2?
8. The first sentence on Page 17 needs to be revised.
9. Typos in Lemma B.2.
10. On Page 19, in the proof of Lemma B.3, why can you use Lemma 4.2 to conclude that $\\|\xi_i\\|\to 0$ as $m\to \infty$? This convergence is in which sense? There should be more explanations.
11. On Page 21, in the equation of $\sum_{i=1}^d(v_i^\top \Phi(X))^2$, why it will converge to 1 if $\Phi$ is an eigenfunction of $\lambda_k$?

**Strengths And Weaknesses:**

## Upsides: ##
This paper is technically solid, focusing on a very well-defined problem for infinite-width or very wide NNs. The written quality of this paper is good. I am able to understand the main point well and the method of proof itself is reasonable and classical. The experiments match the theoretical results of this paper.


## Downsides: ##
1. Modified Spectrum Kernels (MSKs) rely on the spectral decomposition of the neural tangent kernel (NTK) and PGD may rely on the inverse of the NTK, e.g. experiments in Figure 1. However, computing NTK or the empirical NTK for a large sample size is computationally hard. Sometimes, we may need some approximation methods to approximate NTK, like [1-3] below. I think the authors should provide some discussion for the computational issue of the MSK method and what the benefit of this method is for practice.

2. Poor written quality in the proof in the Appendix. I checked the proofs in the Appendix and there are many typos therein (see below as well). Some notions are not clarified at the beginning. For instance, in the Appendix, some norms $\\|\cdot\\|$ are not clarified as operator norm or Frobenius norm at all.

3. The MSK method relies on function $g(\cdot)$, initialization of the training parameters and truncation index $k$. The authors did not provide enough details for how the choose these parameters in practice. The authors should provide more discussions in Section 4.3 and provide more experiments with different $g(\cdot)$, initialization of the training parameters, and truncation index $k$ to present the advantages and limitations of this method.

4. The main theorems should be stated in more detail. In Theorem 4.1 and Lemma 4.2, so far, I do not know how to ensure $\lambda_{\min}(K)>0$ and how this convergence rate for PGD relies on the training sample size $n$ and input feature dimension $d_0$. There are many previous results that proved these properties for GD training, e.g. [4-8]. The authors should include some discussions related to the conditions 1-4 on page 14.

=======================================================================================================

[1] Novak, et al. "Fast finite width neural tangent kernel."

[2] Mohamadi, et al. "A fast, well-founded approximation to the empirical neural tangent kernel."

[3] Engel, Andrew, et al. "Robust Explanations for Deep Neural Networks via Pseudo Neural Tangent Kernel Surrogate Models."

[4] Nguyen, et al. "Tight bounds on the smallest eigenvalue of the neural tangent kernel for deep relu networks."

[5] Montanari and Zhong. "The interpolation phase transition in neural networks: Memorization and generalization under lazy training."

[6] Wang and Zhu. "Deformed semicircle law and concentration of nonlinear random matrices for ultra-wide neural networks."

[7] Oymak and Soltanolkotabi. "Toward moderate overparameterization: Global convergence guarantees for training shallow neural networks."

[8] Wang, et al. "Spectral evolution and invariance in linear-width neural networks."

---

> ### Author Response · Authors · 2023-12-05
> **Response to Reviewer VKSy**
>
> We thank the reviewer for their insightful comments.
>
> - Regarding your comment on the computation of the NTK, we have expanded section 4.3 and now discuss the computational issue and its relation to $k$ in depth.  In particular, we detail how, despite the computational issue, our method is still far more efficient than standard GD or a standard preconditioner. Furthermore, our preconditioner must be computed only once. Hence, its computational complexity is unrelated to the number of iterations needed to converge, which can be exponential as discussed in the article.
>
> - Regarding the function $g$, we have added another experiment comparing various natural choices of $g$, which were not previously discussed in the paper. One of them includes taking $g(\lambda)=\lambda+c$ for some constant $c>0$, implying that $KS=K+cI$. Thus, we obtain a direct correspondence between PGD with such a choice of $g$, and \emph{regularized} kernel regression. Such a choice of $g$ may help prevent overfitting, even without early stopping. We also look at taking $g$ to be constant, or $g(\lambda)=\sqrt{\lambda}$. Interestingly, we see that slower eigenvalue decays help prevent overfitting noise, as suggested by [R3, R4]. We have presented these and lengthened the discussion surrounding $g$.
>
> - We added further explanations elaborating on the role of $k$. In particular, we describe its role in providing computational stability and how this need is magnified since the NTK is computed only approximately. The discussion can be found in section 4.3. Generally, one should consider $k$ as providing a trade-off between the performance of the convergence rate and computational stability and efficiency.
>
> - The initialization of training parameters follows the regular NTK parametrization. Comparisons between the NTK parametrization and a standard one have already been carried out in early NTK works [R2], particularly for the standard FC architecture used in our paper. The experiments from [R2] show that the NTK parametrization performs similarly to a standard one across all the settings they analyzed.
>
> - The statements of the Theorems and the Lemmas are more detailed now. All the assumptions are now clearly stated, and there should be no confusion. Regarding the condition that $\lambda_{min} > 0$, a variety of excellent works, such as [4-8] you referenced, developed a lower bound for the smallest eigenvalue in various settings. We have discussed this in the paper and demonstrated simple conditions for which this holds. Furthermore, the convergence rate does not explicitly depend on the sample size and input feature dimension but only implicitly via the smallest eigenvalue of the NTK matrix. This is characterized by Lemma 4.2.
>
> - The bounded dataset assumption is used for (i) ensuring there is a Mercer decomposition as in Eq 1 and (ii) bounding $f$ at initialization. In both cases, the bounded dataset assumption can be dropped, and the results modified to accommodate other distributions such as Gaussian. We added this discussion to Appendix A and provided more detail there.
>
> - For sufficiently wide networks, $K_t$ is close to $K$ (in spectral norm). Thus, taking $S=K^{-1}$ suffices for achieving the same convergence rates as if $S=K_t^{-1}$. Therefore, we proved our results for a fixed $S$. Within the paragraph you mentioned on page 6, we did not mean that one should take $S=K_t^{-1}$, and we show empirically in Figure 1 that a fixed S works best. However, we do mention that our proofs could be extended to a matrix $S_t$ that varies with $t$. The only condition one would need is to assume that $S_t$ has a Frobenious norm, which is uniformly bounded for all $t$. Because in our proofs $S$ does not vary with $t$, then this condition is trivial. We added these clarifications both in the main text and in the appendix.
>
> - We thank you for the typos you found and requested changes. We have updated the manuscript and included all of these, and we believe that they have significantly improved the clarity of the work. In particular, the norms in question are now clarified. $\lambda_{min}$ is the minimal value of $KS$, and this is now stated in the Lemma. The reference to Lemma 4.2 in the proof of Lemma B.3 was a typo mistake. The correct reference is to Theorem 4.1 and now it is corrected. The equation on page 21 is proved in Braun, 2005 Theorem 4.10, and the discussion which follows clarifies this (see equation 4.15 in their paper). https://d-nb.info/978607309/34
>
> [R1] - Song & Yang, 2019 - Quadratic suffices for over-parametrization via matrix Chernoff bound
>
> [R2] - Lee et al. 2019, Finite Versus Infinite Neural Networks: an Empirical Study
>
> [R3] - Mallinar et al. 2022, Benign, Tempered, or Catastrophic: A Taxonomy of Overfitting
>
> [R4] - Bartlett et al. 2019, Benign Overfitting in Linear Regression
>
> [R5] - Steinwart et al. 2012 - Mercer’s theorem on general domains: On the interaction between measures, kernels, and RKHSs.

---

### Review · Reviewer_UzTC · 2023-11-22

**Summary Of Contributions:**

The authors introduce modified spectrum kernels which can approximate kernels with desired eigenvalues and be used for studying gradient descent methods for training parameters of neural networks.

**Audience:**

Yes

**Broader Impact Concerns:**

The concerns in the previous round have been addressed.

**Claims And Evidence:**

Yes

**Requested Changes:**

Some references should be updated.

**Strengths And Weaknesses:**

The constructed kernels have potential applications in studying neural networks. The realization of the potential could be strengthened for some concrete settings.

---

> ### Author Response · Authors · 2023-12-05
> **Response to Reviewer UzTC**
>
> We thank the reviewer for their feedback. We plan to demonstrate concrete setups and applications in future work. Such settings may include image reconstruction or rendering, and Physics Informed Neural Networks (PINN), which were shown to suffer significantly from the slow convergence rate in the directions corresponding to small eigenvalues of the NTK matrix [R1].
> The references have been updated; thank you for your suggestion.
>
> [R1] - Wang et al. 2022 - When and why PINNs fail to train: A neural tangent kernel perspective

---

### Decision · Action_Editor_eZbv · 2024-01-18

**Recommendation:** Accept with minor revision

**Comment:**

The authors have already made several improvements and all reviewers vote for either Accept or Leaning Accept. One reviewer mentions
"..., I am not convinced by the author's explanation of computing efficiency. They claim their method still results in more efficient optimization in the over-parameterized regime by comparing with the NTK results from (Song & Yang, 2019) and showing that inverting K or calculating its eigenvalues is cheaper than computing the gradient. But Song & Yang did not obtain the optimal width for NTK theory. I do not think this is a convincing evidence."
I suggest a minor revision, where the authors either provide more evidence for their claim, or appropriately qualify their claim (e.g. an improved sentence or paragraph along the lines of "our method results in more efficient optimization in the over-parameterized regime comparing with known methodology relying on the NTK (Song & Yang, 2019), however we note that Song & Yang were not able to obtain the optimal width, ..."

**Audience:**

This paper is of interest to many of TMLR's audience. The authors describe a method for controlling inductive bias of networks with a modified kernel that approximates kernels with target eigenvalues, which is interesting from a methodological perspective. Using the machinery of the NTK is also interesting from a technical perspective.

**Claims And Evidence:**

All reviewers agree that claims are supported by evidence. Reviewer VKSy pointed out several minor errors and typos, and these were addressed by the authors. The authors also clarified several points raised by the reviewer in their updated manuscript. Authors also engaged in a discussion with Reviewer wmuT to improve the quality of illustrations in the paper.

---

> ### Author Response · Authors · 2024-02-28
> **Reply to Decision**
>
> We are very happy to hear of the decision and have uploaded the camera-ready version. We have revised the paragraph that discusses the efficiency of our method and have significantly strengthened our argument. Furthermore, when mentioning the bounds of Song & Yang, we clarified that their bounds may not be optimal.